# Rostral and caudal basolateral amygdala engage distinct circuits in the prelimbic and infralimbic prefrontal cortex

**Kasra Manoocheri, Adam G Carter\***

Center for Neural Science, New York University, New York, United States

**Abstract** Connections from the basolateral amygdala (BLA) to medial prefrontal cortex (PFC) regulate memory and emotion and become disrupted in neuropsychiatric disorders. The diverse roles attributed to interactions between the BLA and PFC may reflect multiple circuits nested within a wider network. To examine these circuits, we first used retrograde and anterograde anatomy to show that the rostral BLA (rBLA) and caudal BLA (cBLA) differentially project to prelimbic (PL) and infralimbic (IL) subregions of the mouse PFC. Using ex vivo whole-cell recordings and optogenetics, we then assessed which neuronal subtypes are targeted, showing that rBLA preferentially drives layer 2 (L2) cortico-amygdalar (CA) neurons in PL, whereas cBLA drives layer 5 (L5) pyramidal tract (PT) neurons in IL. We next combined in vivo silicon probe recordings and optogenetics to confirm that cBLA mainly influences IL L5, whereas rBLA primarily activates PL L2, but also evokes polysynaptic activity in PL L5. Lastly, we used soma-tagged optogenetics to explore the local circuits linking superficial and deep layers of PL, showing how rBLA can engage L2 CA neurons to impact L5 PT neuron activity. Together, our findings delineate how subregions of the BLA target distinct networks within the PFC and differentially influence output from PL and IL.

**\*For correspondence:**
adam.carter@nyu.edu

**Competing interest:** The authors declare that no competing interests exist.

## Editor's evaluation

This paper will be of interest to readers studying the neuronal circuit in general and those studying the basolateral amygdala (BLA), prefrontal cortex (PFC), and diseases associated with these regions. Using innovative circuit analysis techniques, this important paper shows that different subregions of the BLA form cell-type-specific connections with the subregion of PFC and engage specific local circuits within it. The key claim of the paper is supported by compelling data from multiple approaches.

## Introduction

Connections from the basolateral amygdala (BLA) to the medial prefrontal cortex (PFC) regulate memory and emotion (*Phelps and LeDoux, 2005*; *Etkin et al., 2011*). Dysfunction of these connections contributes to neuropsychiatric disorders, including post-traumatic stress disorder, depression, and autism (*Gilboa et al., 2004*; *Felix-Ortiz et al., 2016*; *McTeague et al., 2020*). The BLA sends strong excitatory inputs to the PFC, which contact specific cell types to drive local activity and output to downstream brain areas (*Sotres-Bayon et al., 2012*; *Little and Carter, 2013*; *Cheriyan et al., 2016*; *Burgos-Robles et al., 2017*). However, previous studies have treated the BLA and PFC as monolithic entities, whereas each structure contains multiple subregions that may have distinct connections and functional roles (*McDonald, 1991*; *Hintiryan et al., 2021*).

The PFC is subdivided along the dorsoventral axis into the prelimbic (PL) and infralimbic (IL) regions (*Van De Werd et al., 2010*; *Van De Werd and Uylings, 2014*), which play different functional roles

in behavioral paradigms like threat conditioning (*Sierra-Mercado et al., 2011*). Similarly, the BLA is subdivided along the rostro-caudal axis into anterior (magnocellular) and posterior (parvocellular) regions (*McDonald, 2003*; *O'Leary et al., 2020*), which also play distinct roles in aversive and appetitive behaviors (*Kim et al., 2017*; *Zhang et al., 2021*). Anatomical studies using transgenic mice suggest one population of cells in anterior BLA targets PL, while a separate group in posterior BLA targets IL (*Kim et al., 2016*). However, recent work suggests a more complicated relationship, with multiple projections from cells distributed along the rostral to caudal axis of the BLA (*Hintiryan et al., 2021*), which have an unknown influence on activity and output of the PFC.

In addition to contacting different subregions and layers, long-range inputs often make targeted connections onto specific cell types in the PFC (*Anastasiades and Carter, 2021*). Recent studies suggest that glutamatergic projections from the BLA may differentially activate specific networks in PL compared to IL. BLA inputs to PL target layer 2 (L2) cortico-amygdalar (CA) neurons, which in turn project back to the BLA, forming a direct, reciprocal circuit under the control of local inhibition (*Little and Carter, 2013*; *McGarry and Carter, 2016*). In contrast, BLA connections in IL can activate layer 5 (L5) pyramidal tract (PT) neurons, which project to the periaqueductal gray (PAG) and other brain regions to exert top-down control of behavior (*Cheriyan et al., 2016*). We hypothesize that these opposing findings reflect distinct connections from rostral BLA (rBLA) and caudal BLA (cBLA) onto specific cell types within PL and IL, reflecting two parallel networks that link these brain regions.

In addition to their direct connections, long-range inputs can evoke polysynaptic excitation and inhibition in the PFC (*Collins et al., 2018*; *Huang et al., 2019*). Connections within and across layers are understudied in PFC, but well explored in other cortices that lack layer 4 (L4) (*Hooks et al., 2011*). In motor cortex, layer 2/3 (L2/3) neurons project to deeper layers and target L5 pyramidal neurons (*Anderson et al., 2010*; *Hirai et al., 2012*), whereas L5 PT neurons predominantly target nearby L5 PT neurons (*Morishima and Kawaguchi, 2006*). Similarly, inhibition mediated by local interneurons is transmitted within and across layers (*Kätzel et al., 2011*; *Saffari et al., 2016*; *Tremblay et al., 2016*). By activating specific cell types in different layers, the BLA can evoke complex polysynaptic activity, but it is unknown how rBLA and cBLA inputs engage these circuits.

Here, we examine how the rBLA and cBLA target distinct subregions, layers, and cell types in the mouse PL and IL. We first use anatomy to show that rBLA primarily projects to PL, and cBLA mostly projects to IL, indicating distinct connections. We next use ex vivo recordings and optogenetics to assess cell type-specific connections, showing that rBLA preferentially activates PL L2 CA neurons and cBLA engages IL L5 PT neurons. We then combine in vivo recordings with optogenetics and find that while cBLA influences are restricted to IL L5, rBLA drives both PL L2 and evokes robust polysynaptic activity in PL L5. Lastly, using soma-tagged optogenetics, we establish that this local influence reflects connections from PL L2 CA neurons onto PL L5 PT neurons within PL. Together, our findings illustrate how rBLA and cBLA evoke activity in PL and IL, demonstrating how subregions of the BLA engage distinct cortical networks in the PFC.

## Results
### Distinct projections of rostral and caudal BLA to PL and IL PFC

To confirm if subregions of the PFC receive distinct and spatially segregated inputs from the BLA, we first injected retrograde viruses (AAVrg-tdTomato and AAVrg-GFP) into both PL and IL of the same animals, at offset anterior–posterior coordinates to minimize potential overlap (*Figure 1A*). We observed largely separate populations of PL and IL-projecting neurons across the BLA, with a smaller population of dual-projection neurons (PL = 583 ± 59 cells, IL = 694 ± 78 cells, dual = 246 ± 30 cells, *n* = 3 animals) (*Figure 1B*). Labeled cells were distributed along a gradient in the rostro-caudal axis, with PL-projecting neurons biased toward the rostral BLA (rBLA), and IL-projecting neurons shifted toward the cBLA (*Figure 1C*). We also found a similar gradient in experiments in which we injected AAVrg-tdTomato into PL or IL at the same anterior–posterior coordinate (*Figure 1—figure supplement 1*). These results agree with studies using transgenic lines (*Kim et al., 2016*), and suggest PL and IL receive inputs from neurons residing in rBLA and cBLA, respectively.

To directly compare rBLA and cBLA projections to PFC, we next injected anterograde viruses (AAV-ChR2-eYFP and AAV-ChR2-mCherry) into rBLA and cBLA (−1.1 and −1.7 mm from bregma) in the same animals (*n* = 3) (*Figure 1D*). Care was taken to avoid injecting in the ventral hippocampus, which

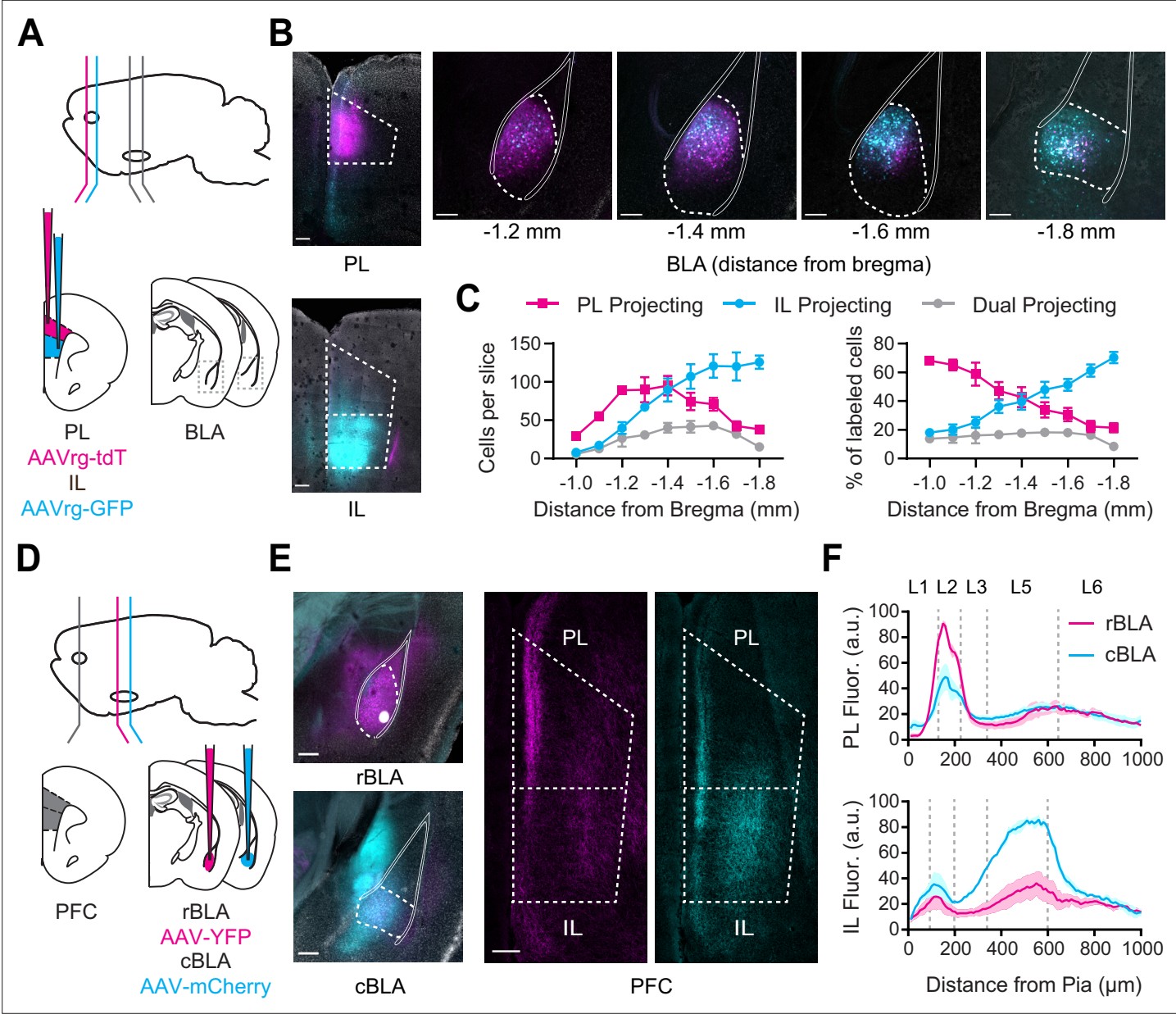

**Figure 1.** Rostral basolateral amygdala (rBLA) and caudal basolateral amygdala (cBLA) project to different subregions and layers of the prefrontal cortex (PFC). (**A**) Schematic for injections of AAVrg-tdTomato into prelimbic (PL) (magenta) and AAVrg-GFP into infralimbic (IL) (cyan). (**B**) *Left*, Images of injection sites in PL (+2.4 mm from bregma, magenta) and IL (+2.0 mm from bregma, cyan). Scale bar = 200 μm. *Right*, Images of retrogradely labeled PL-projecting (magenta) and IL-projecting neurons (cyan) across the AP axis of basolateral amygdala (BLA) (distances relative to bregma). Solid white lines mark white matter boundaries of the BLA. Scale bar = 200 μm. 4',6-diamidino-2-phenylindole (DAPI) staining in gray. (**C**) *Left*, Average number of PL-projecting, IL-projecting, or dual-projecting cells along AP axis of BLA. *Right*, Percentage of PL-projecting, IL-projecting, and dual-projecting cells in the BLA (*n* = 3 animals). (**D**) Schematic for injections of AAV-YFP into rBLA (−1.1 mm from bregma; magenta) and AAV-mCherry into cBLA (−1.7 mm from bregma; cyan). (**E**) *Left*, Images of injection sites in rBLA (magenta) and cBLA (cyan). Scale bar = 250 μm. DAPI staining in gray. *Right*, Images of PL and IL (+2.2 mm from bregma), showing anterogradely labeled rBLA (magenta) and cBLA (cyan) axonal projections. Scale bar = 200 μm. (**F**) *Top*, Summary of fluorescence intensity from pia to white matter of rBLA (magenta) and cBLA (cyan) projections to PL, normalized to maximal fluorescence in each slice. *Bottom*, Similar for rBLA and cBLA projections to IL (*n* = 3 animals). See also *Figure 1—figure supplements 1 and 2*.

The online version of this article includes the following figure supplement(s) for figure 1:

**Figure supplement 1.** Prelimbic (PL) and infralimbic (IL)-projecting neurons across the basolateral amygdala (BLA).

**Figure supplement 2.** Rostral basolateral amygdala (rBLA) projections to prelimbic (PL) and caudal basolateral amygdala (cBLA) projections to infralimbic (IL).

abuts the cBLA and also projects to the PFC (*Liu and Carter, 2018*; *Liu et al., 2020*). Consistent with our retrograde labeling, we observed rBLA axons primarily in PL and cBLA axons in IL, with distinct laminar targeting in each subregion (*Figure 1E*). Quantifying axon fluorescence across layers and subregions established that rBLA primarily targeted PL L2, whereas cBLA targeted IL L5 (*Figure 1F*). Together, these data indicate that rostral and caudal BLA form distinct anatomical connections across layers and the dorsal–ventral axis of the PFC.

Lastly, to establish the presence of separate rBLA to PL and cBLA to IL projections, we next injected a combination of retrograde AAVrg-Cre into either PL or IL, followed by anterograde AAV-DIO-YFP into rBLA or cBLA, respectively, and quantified axonal fluorescence across layers and subregions of PFC (*Figure 1—figure supplement 2*). We again found that rBLA axons were prominent in PL L2 while cBLA axons were strongest in IL L5, consistent with our other anatomy and establishing that PL-projecting neurons in rBLA and IL-projecting neurons in cBLA do not collateralize within PFC.

## Rostral and caudal BLA target different cell types in PL and IL PFC

Our anatomy showed that rBLA and cBLA project to different subregions and layers of PFC, but did not reveal which cell types are targeted within them. Previous work has shown that BLA inputs can selectively target L2 CA neurons in PL or L5 PT neurons in IL (*Little and Carter, 2013*; *Cheriyan et al., 2016*; *McGarry and Carter, 2016*). We next used whole-cell recordings and optogenetics to test whether rBLA and cBLA are responsible for engaging these different cell types in the PFC. We injected ChR2-expressing virus (AAV-ChR2-EYFP) into either rBLA or cBLA to visualize and stimulate axons in the PFC (*Figure 2A*; *Little and Carter, 2013*; *McGarry and Carter, 2016*). In the same mice, we also coinjected retrogradely transported, fluorescently tagged cholera toxin subunit B (CTB) into either rBLA or cBLA and the PAG to label CA and PT neurons, respectively (*Figure 2A*; *Ferreira et al., 2015*; *Collins et al., 2018*; *Liu and Carter, 2018*). After waiting for expression and transport, we observed CA neuron labeling in L2 and PT neuron labeling in L5 across PL and IL (*Figure 2A*, *Figure 2—figure supplement 1*). We then used wide-field illumination to activate rBLA or cBLA inputs, and measured synaptic responses at CA and PT neurons in TTX (1 µM), 4-AP (100 µM) and high extracellular Ca$^{2+}$ (4 mM) to isolate monosynaptic connections (*Petreanu et al., 2009*; *Mao et al., 2011*). To control for variability in virus expression between slices and animals, we compared the ratio of responses referenced to a specific cell type (either PL L2 CA for rBLA or IL L5 PT for cBLA) (*Figure 2B*).

Taking this approach, we observed that rBLA-evoked excitatory postsynaptic currents (EPSCs) at −60 mV were much larger at PL L2 CA neurons compared to all other cell types in the different layers and subregions (EPSC ratio: IL L2 CA vs. PL L2 CA = 0.03, geometric standard deviation factor [GSD] = 12.3, p = 0.0003; n = 8 pairs; PL L5 PT vs. PL L2 CA = 0.02, GSD = 2.6, p = 0.0001; n = 8 pairs; IL L5 PT vs. PL L2 CA = 0.09, GSD = 7.7, p = 0.006; n = 8 pairs; 5 animals) (*Figure 2C*, *Figure 2—figure supplement 1*). In contrast, cBLA-evoked EPSCs were much larger at IL L5 PT neurons compared to all other cell types (EPSC ratio: PL L2 CA vs. IL L5 PT = 0.06, GSD = 9.3, p = 0.005; n = 7 pairs; IL L2 CA vs. IL L5 PT = 0.11, GSD = 10.5, p = 0.014; n = 7 pairs; PL L5 PT vs. IL L5 PT = 0.008, GSD = 3.1, p < 0.0001; n = 8 pairs; 6 animals) (*Figure 2D*, *Figure 2—figure supplement 1*). In separate experiments, we also confirmed that rBLA inputs target PL L2 CA neurons over L2 cortico-cortical (CC) neurons and IL L5 CA neurons (*Figure 2—figure supplement 1*) and that cBLA inputs target IL L5 PT over IL L5 CA neurons (*Figure 2—figure supplement 1*), consistent with previous results on non-specific BLA inputs. These findings show that subregions of BLA preferentially target different layers and cell types in the PFC, and further indicate that rBLA primarily synapses onto PL L2 CA neurons, and cBLA primarily contacts IL L5 PT neurons.

Other long-range excitatory inputs to the PFC can evoke unique responses by selectively targeting the apical dendrites of pyramidal neurons (*Anastasiades et al., 2021*). Because our anatomy showed prominent BLA axons in superficial layers, we also tested the possibility that BLA inputs contact the apical dendrites of L5 PT neurons using subcellular Channelrhodopsin-Assisted Circuit Mapping (sCRACM) in the presence of TTX and 4-AP (*Petreanu et al., 2009*), stimulating across a grid aligned to the pia and soma (*Anastasiades et al., 2021*; *Figure 2E*). We found strong rBLA and cBLA input to the basal and perisomatic dendrites of PL L2 CA and IL L5 PT neurons, respectively, and no rBLA input to either the apical or basal dendrites of PL L5 PT neurons (*Figure 2F*, *Figure 2—figure supplement 1*). Together, these findings indicate that rBLA and cBLA primarily make connections close to the soma of pyramidal neurons in the PFC, and do not target the apical dendrites of deep layer neurons.

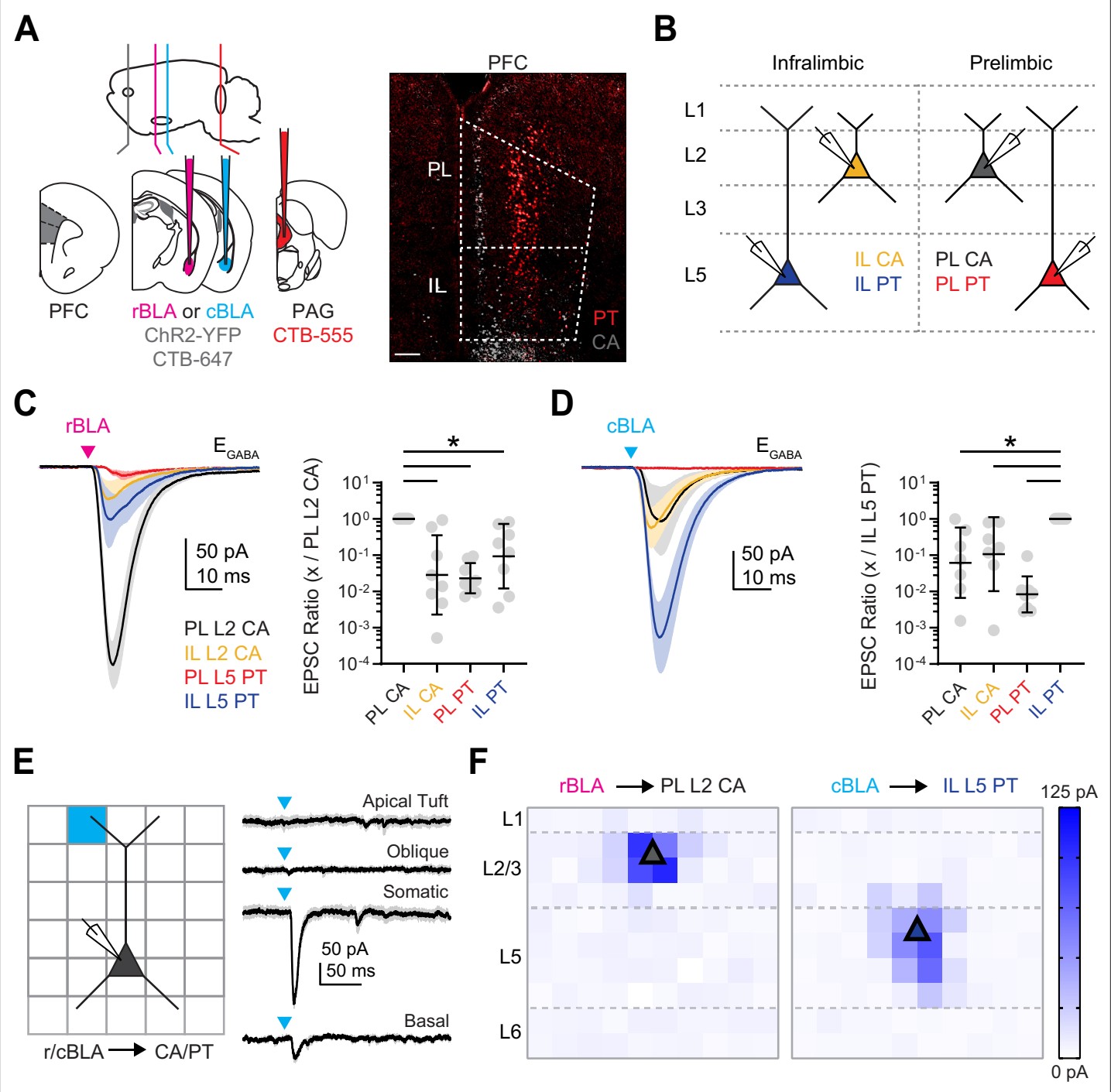

**Figure 2.** Rostral basolateral amygdala (rBLA) targets prelimbic (PL) L2 cortico-amygdalar (CA) neurons and caudal basolateral amygdala (cBLA) targets infralimbic (IL) L5 pyramidal tract (PT) neurons. (**A**) *Left*, Schematic for injections of AAV-ChR2-eYFP and CTB-647 (gray) into rBLA or cBLA and CTB-555 (red) into periaqueductal gray (PAG). *Right*, Example image of retrogradely labeled CA neurons (gray) and PT neurons (red) in prefrontal cortex (PFC). Scale bar = 200 μm. (**B**) Schematic of color-coded PL L2 CA, IL L2 CA, PL L5 PT, and IL L5 PT neurons recorded in the presence of TTX and 4-AP to isolate monosynaptic connections from either rBLA or cBLA. (**C**) *Left*, Average rBLA-evoked excitatory postsynaptic currents (EPSCs) recorded at L2 CA and L5 PT neurons in PL and IL. Magenta arrow = light stimulation. *Right*, Summary of EPSC amplitude ratios, comparing responses at PL L2 CA neurons with paired IL L2 CA (*n* = 8), IL L5 PT (*n* = 8), and PL L5 PT (*n* = 8) neurons (*n* = 5 animals). (**D**) Similar to (**C**) for cBLA-evoked EPSCs, comparing responses at IL L5 PT neurons with paired IL L2 CA (*n* = 7), PL L2 CA (*n* = 7), and PL L5 PT (*n* = 8) neurons (*n* = 6 animals). (**E**) *Left*, Schematic for sCRACM experiments, showing pseudorandom illumination (blue light) of 75 × 75 μm squares across L1 to L6 of PFC, while recording either rBLA-evoked EPSCs at PL L2 CA neurons or cBLA-evoked EPSCs and IL L5 PT neurons in the presence of TTX and 4-AP. *Right*, Example cBLA-evoked EPSCs recorded at a IL

*Figure 2 continued on next page*

*Figure 2 continued*

L5 PT neuron, including with stimulation at apical tuft, oblique, somatic, and basal dendrite locations. (**F**) *Left*, Average rBLA-evoked EPSC maps for PL L2 CA neurons (*n* = 4 cells, 2 animals), color coded by amplitude. *Right*, Average cBLA-evoked EPSC maps for IL L5 PT neurons (*n* = 6 cells, 4 animals). Triangle denotes typical cell body location. *p < 0.05. See also *Figure 2—figure supplement 1*.

The online version of this article includes the following figure supplement(s) for figure 2:

**Figure supplement 1.** Rostral basolateral amygdala (rBLA) and caudal basolateral amygdala (cBLA) monosynaptic inputs.

## Rostral and caudal BLA activate different networks in PL and IL PFC

Having established which cell types are targeted by rBLA and cBLA, we next assessed how they are engaged during more physiological conditions and activity patterns. We stimulated rBLA or cBLA inputs with the repetitive trains (5 pulses of 473 nm light for 2 ms at 20 Hz) and recorded EPSCs at −60 mV ($E_{GABA}$) and inhibitory postsynaptic currents (IPSCs) at +15 mV ($E_{AMPA}$) in the presence of CPP (10 µM) to block N-methyl-D-aspartate (NMDA) receptors. For rBLA inputs, we recorded pairs of PL L2 CA neurons and IL L5 PT neurons (*Figure 3A*). We found that rBLA-evoked EPSCs were again largest at PL L2 CA neurons ($EPSC_1$: PL L2 CA = −401 ± 63 pA; IL L5 PT = −133 ± 55 pA; PL L2 CA vs. IL L5 PT, p = 0.0078, *n* = 8 pairs, 5 animals) (*Figure 3B*). In contrast, rBLA-evoked IPSCs were of comparable amplitude at PL L2 CA and IL L5 PT neurons ($IPSC_1$: PL L2 CA = 450 ± 86 pA; IL L5 PT = 521 ± 136 pA; PL L2 CA vs. IL L5 PT, p = 0.8125, *n* = 7 pairs, 5 animals) (*Figure 3B*). We found differences in excitation/inhibition (*E/I*) ratios, with rBLA inputs onto PL L2 CA neurons showing higher values than IL L5 PT neurons (rBLA *E/I* ratio: PL L2 CA = 0.99, GSD = 1.9; IL L5 PT = 0.15, GSD = 2.7; PL L2 CA vs. IL L5 PT, p = 0.0156, *n* = 7 pairs, 5 animals) (*Figure 3B*). For cBLA inputs, we recorded pairs of IL L5 PT neurons and PL L2 CA neurons (*Figure 3C*). We found that both EPSCs and IPSCs were strongly biased onto IL L5 PT neurons ($EPSC_1$: PL L2 CA = −31 ± 10 pA; IL L5 PT = −415 ± 139 pA; PL L2 CA vs. IL L5 PT, p = 0.0156, *n* = 7 pairs, 4 animals) ($IPSC_1$: PL L2 CA = 31 ± 28 pA; IL L5 PT = 858 ± 149 pA; PL L2 CA vs. IL L5 PT, p = 0.0156, *n* = 7 pairs, 4 animals) (cBLA *E/I* ratio; IL L5 PT = 0.4, GSD = 2.3; *n* = 7 pairs, 4 animals) (*Figure 3D*). In a subset of recordings, we also recorded from PL L5 PT and IL L2 CA neurons in addition to PL L2 CA or IL L5 PT neurons and found similar targeting biases from rBLA and cBLA, with the exception of prominent inhibition on PL L5 PT neurons evoked by rBLA stimulation (*Figure 3—figure supplement 1*). These results show that rBLA and cBLA directly excite their targets and evoke feed-forward inhibition that could prevent firing of action potentials (APs).

We then tested how rBLA and cBLA inputs drive cell type-specific firing of the main recipient cell types in PL and IL. Projection neurons in the PFC have distinct intrinsic properties (*Ferreira et al., 2015*), with CA neurons resting at very hyperpolarized potentials (*McGarry and Carter, 2016*), and PT neurons displaying robust h-current (*Dembrow et al., 2010*; *Anastasiades et al., 2018*). Taking these properties into account, we recorded responses in current-clamp at resting membrane potential (RMP). We found that rBLA inputs evoked significantly more action potential firing of PL L2 CA neurons compared to IL L5 PT neurons in the same slices (firing probability: PL L2 CA = 0.88 ± 0.05, IL L5 PT = 0.23 ± 0.15, p = 0.016; *n* = 8 pairs, 5 animals) (*Figure 3E*). In contrast, cBLA inputs activated only IL L5 PT neurons and never activated PL L2 CA neurons (firing probability: PL L2 CA = 0, IL L5 PT = 0.98 ± 0.02, p = 0.03; *n* = 7 pairs, 3 animals) (*Figure 3F*). Together, these findings indicate rBLA and cBLA engage distinct projection neuron networks in the PFC, supporting the hypothesis that BLA consists of functionally distinct rostral and caudal divisions.

## Rostral and caudal BLA evoke distinct activity in PL and IL PFC

To test these predictions, and assess differences in the impact of rBLA and cBLA inputs on the PFC, we next combined in vivo optogenetics and Neuropixels (NP) recordings (*Jun et al., 2017*). We first injected AAV-ChR2-EYFP into either rBLA or cBLA and implanted an optical fiber above the injection site (*Figure 4A*). After waiting for expression, we acutely recorded light-evoked activity in the PFC of awake, head-fixed mice. The thin dimensions and high density of NP probes allowed us to target L2 or L5 of PL and IL in the same animals (*Figure 4B*). We confirmed the locations of each NP probe by registering to the Allen common coordinate framework (CCF) (*Wang et al., 2020*; *Figure 4—figure supplement 1*). Our stimulus protocol (5 × 2 ms pulses @ 20 Hz) was similar to previous studies on BLA to PFC connections (*Floresco and Tse, 2007*; *Felix-Ortiz et al., 2016*). Light-evoked activity was

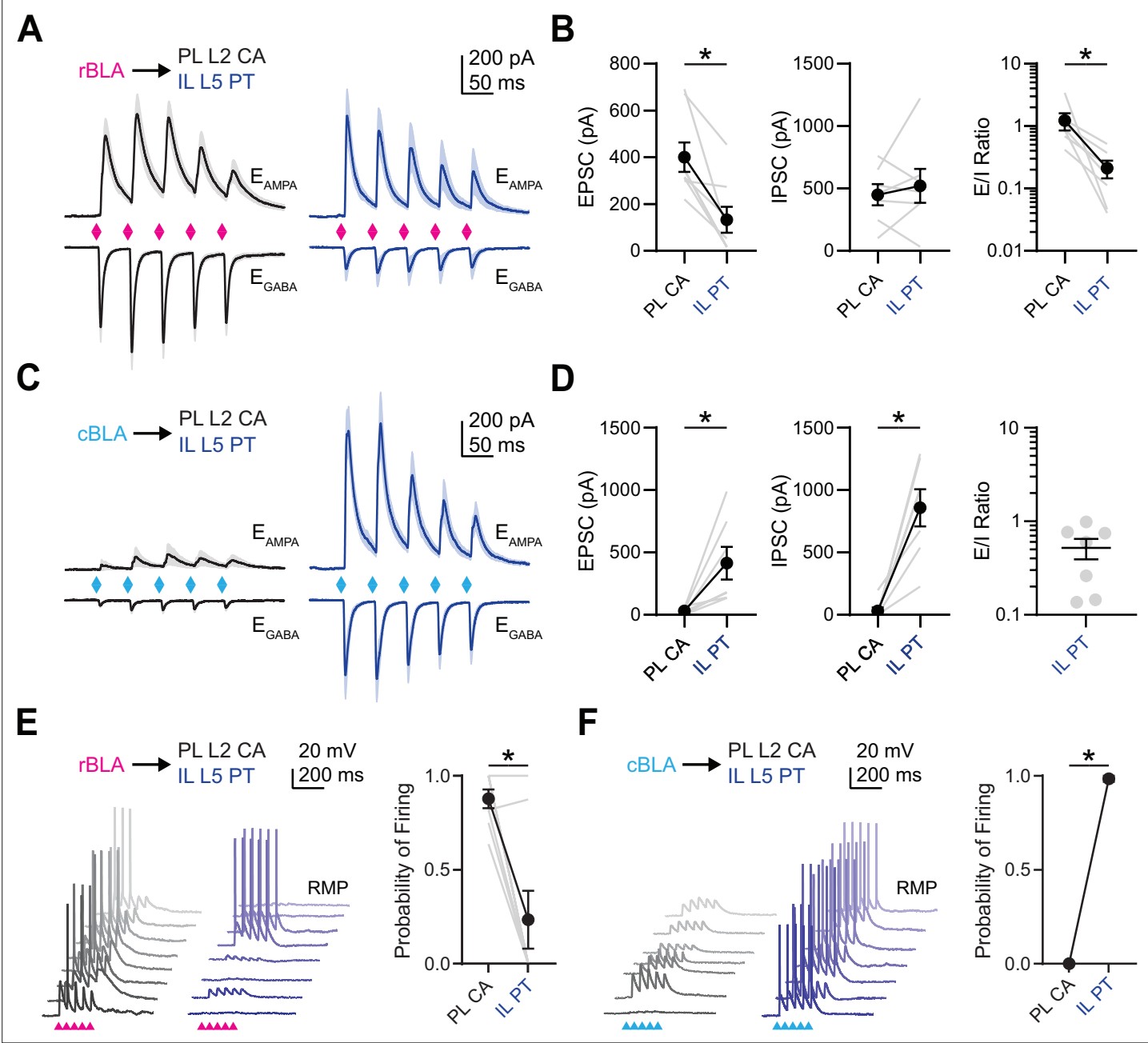

**Figure 3.** Rostral basolateral amygdala (rBLA) and caudal basolateral amygdala (cBLA) activate distinct projection neurons in prelimbic (PL) and infralimbic (IL). (**A**) Average rBLA-evoked excitatory postsynaptic currents (EPSCs) (bottom, recorded at $E_{GABA}$) and IPSCs (top, recorded at $E_{AMPA}$) at PL L2 cortico-amygdalar (CA) (left, black) and IL L5 pyramidal tract (PT) neurons (right, blue) ($n$ = 8 pairs, 5 animals). Diamonds denote light pulses. (**B**) *Left*, Summary of rBLA-evoked $EPSC_1$ amplitudes for PL L2 CA and IL L5 PT neurons. *Middle*, Similar for $IPSC_1$ amplitudes. *Right*, Similar for $EPSC_1/IPSC_1$ ratios. Gray lines denote individual pairs of neurons. (**C, D**) Similar to (A, B) for cBLA inputs onto PL L2 CA and IL L5 PT neurons, with $E/I$ ratios only for IL L5 PT due to the lack of inhibition onto PL L2 CA ($n$ = 7 pairs, 4 animals). (**E**) *Left*, Example rBLA-evoked EPSPs and APs from pairs of PL L2 CA and IL L5 PT neurons recorded in current-clamp at resting membrane potential (RMP). *Right*, Summary of rBLA-evoked AP probability for PL L2 CA and IL L5 PT neurons. Gray lines denote individual pairs of neurons ($n$ = 8 pairs, 5 animals). (**F**) Similar to (**E**) for cBLA-evoked EPSPs and APs from pairs of PL L2 CA and IL L5 PT neurons ($n$ = 7 pairs, 3 animals). Note that all gray lines overlap with the black line. *$p < 0.05$. See also *Figure 3—figure supplement 1*.

The online version of this article includes the following figure supplement(s) for figure 3:

**Figure supplement 1.** Short-term dynamics of rostral basolateral amygdala (rBLA) and caudal basolateral amygdala (cBLA) inputs.

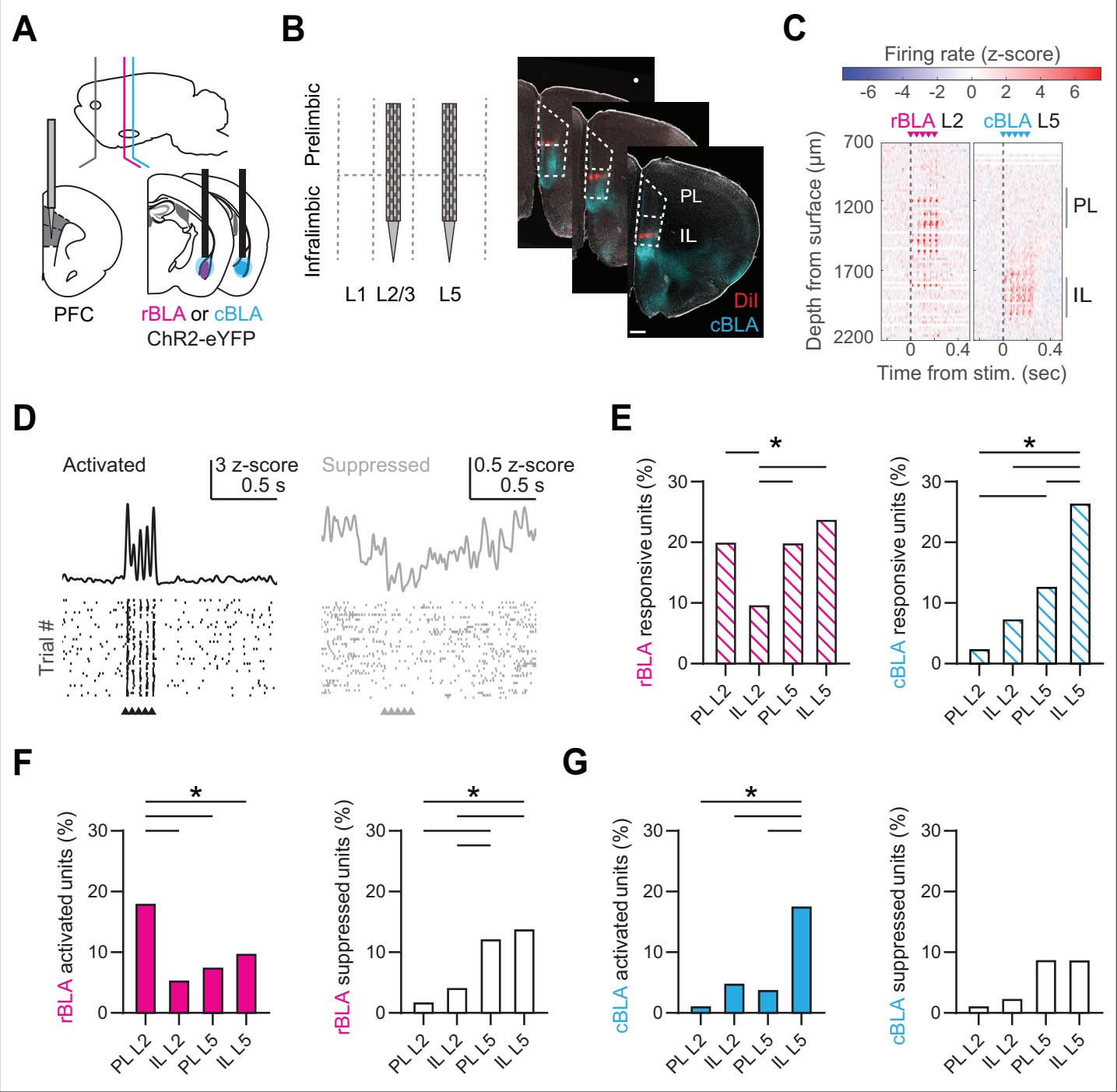

**Figure 4.** Rostral basolateral amygdala (rBLA) and caudal basolateral amygdala (cBLA) stimulation evoke distinct responses in prelimbic (PL) and infralimbic (IL). (**A**) Schematic of AAV-ChR2-eYFP injections and fiber optic placement into either rBLA (magenta) or cBLA (cyan), along with Neuropixels (NP) recordings in prefrontal cortex (PFC). (**B**) *Left*, Schematic of NP probe insertions into L2 and L5 of IL and PL. *Right*, Example of NP probe tracts from sequential insertions into L2 and L5 of PFC (red = DiI), along with cBLA axons (cyan = eYFP). DAPI staining is shown in gray. Scale bar = 500 μm. (**C**) *Left*, Example of rBLA stimulation and NP recording in L2 of PFC. Graph shows the average change in spiking activity (*z*-scored) evoked by rBLA stimulation across each channel of the NP probe. Arrows denote timing of light-emitting diode (LED) stimulation. *Right*, Similar for cBLA stimulation and NP recording in L5 of PFC. (**D**) *Left*, Example single unit with significant basolateral amygdala (BLA)-evoked activation, with average response (top) and raster plot of peri-stimulus activity for 40 trials (bottom). *Right*, Similar for single unit with significant BLA-evoked suppression. (**E**) Percentage of units responsive to rBLA (left) or cBLA (right) inputs across different layers and subregions of the PFC. (**F**) Similar to (**E**) for units activated (left) or suppressed (right) by rBLA inputs. (**G**) Similar to (**E**) for units activated (left) or suppressed (right) by cBLA inputs. *p < 0.05. See also *Figure 4—figure supplements 1 and 2*.

*Figure 4 continued on next page*

*Figure 4 continued*

The online version of this article includes the following figure supplement(s) for figure 4:

**Figure supplement 1.** Rostral basolateral amygdala (rBLA) and caudal basolateral amygdala (cBLA) inputs cause distinct patterns of activity across prefrontal cortex (PFC).

**Figure supplement 2.** Rostral basolateral amygdala (rBLA) and caudal basolateral amygdala (cBLA) excited and inhibited units.

recorded across the dorsal–ventral axis of the PFC and analyzed from NP probes located in either L2 (*n* = 9 insertions from 6 mice) or L5 (*n* = 8 insertions from 6 mice) of PL and IL (*Figure 4C*).

To characterize BLA-evoked activity, we used semi-automated spike-sorting to distinguish single units (rBLA-evoked recordings = 979 single units/1897 total; cBLA-evoked recordings = 596 single units/1152 total). We found that rBLA and cBLA evoked a variety of responses across the PFC, leading to significantly activated or suppressed units (*Figure 4D*). rBLA stimulation led to more responsive units in PL L2, PL L5, and IL L5 compared to IL L2 (PL L2 = 20.1%; IL L2 = 9.8%; PL L5 = 20.0%; IL L5 = 23.9% of single units in subregion), whereas cBLA stimulation mainly drove responses in IL L5 (PL L2 = 2.6%; IL L2 = 7.5%; PL L5 = 12.9%; IL L5 = 26.6% of single units in subregion) (*Figure 4E*). The highest proportion of rBLA-activated units were found in PL L2 (PL L2 = 18.2%; IL L2 = 5.5%; PL L5 = 7.9%; IL L5 = 9.9% of single units in subregion), whereas rBLA-suppressed units were enriched in L5 of PL and IL (PL L2 = 1.9%; IL L2 = 4.3%; PL L5 = 12.4%; IL L5 = 14% of single units in subregion) (*Figure 4F*, *Figure 4—figure supplement 2*). In contrast, cBLA-activated units were greatest in IL L5 (PL L2 = 1.3%; IL L2 = 5%; PL L5 = 4%; IL L5 = 17.7% of single units in subregion), with no significant differences in the proportion of cBLA-suppressed units (PL L2 = 1.3%; IL L2 = 2.5%; PL L5 = 8.9%; IL L5 = 8.9% of single units in subregion) (*Figure 4G*, *Figure 4—figure supplement 2*). We observed no significant differences in latency to spike after each light-emitting diode (LED) pulse in activated units (*Figure 4—figure supplement 2*). Together, these findings indicate that rBLA and cBLA drive distinct activity in PL and IL, with rBLA primarily activating PL L2, but also unexpectedly generating mixed activation and suppression across L5, and cBLA predominantly activating IL L5, with few responses elsewhere.

## Rostral BLA evokes polysynaptic excitation and inhibition of PL L5 PT neurons

While our in vivo recordings indicate that rBLA inputs primarily target PL L2, we also observed a substantial number of activated and suppressed units in PL L5 and IL L5. Previous studies also found that BLA can influence the firing of PL L5 PT neurons to shape PFC output (*Huang et al., 2019*). Because we found no monosynaptic connections onto PL L5 PT neurons, we hypothesized a role for local connections. To test this idea, we first combined current-clamp recordings of PL L2 CA neurons with voltage-clamp recordings of PL L5 PT neurons (*Figure 5A*). We illuminated rBLA inputs to PL L2 with trains that evoked subthreshold or suprathreshold activity at PL L2 CA neurons. In the same slices, we recorded rBLA-evoked EPSCs at $E_{GABA}$ and IPSCs at $E_{AMPA}$ from PL L5 PT neurons. We found no responses at subthreshold intensities, but robust responses at suprathreshold intensities, suggesting polysynaptic activity from L2 to L5 is involved in rBLA-evoked responses at PL L5 PT neurons (EPSC: sub = −9 ± 4 pA, supra = −113 ± 24 pA, p = 0.02; *n* = 8 pairs; IPSC: sub = 17 ± 5 pA, supra = 508 ± 101 pA, p = 0.008; *n* = 8 pairs; 4 animals) (*Figure 5B, C*). In contrast, we observed no responses in IL L5 PT neurons at either subthreshold or suprathreshold intensities, suggesting activity does not spread from PL L2 to IL L5 in the slice, and that rBLA instead projects directly to IL L5 (EPSC: sub = −4 ± 3 pA, supra = −7 ± 4 pA, p = 0.055; *n* = 8 pairs; IPSC: sub = −7 ± 2 pA, supra = −11 ± 3 pA, p = 0.64; *n* = 8 pairs; 3 animals) (*Figure 5D–F*). In order to confirm that cBLA does not engage PL L5 via IL L5, we also activated cBLA inputs in IL L5, and observed no EPSCs in PL L5 PT neurons at either subthreshold or suprathreshold intensities, and only minimal IPSCs at suprathreshold intensities, suggesting polysynaptic activity also does not spread from IL L5 to PL L5 (EPSC: sub = −4 ± 2 pA, supra = −13 ± 5 pA, p = 0.055; *n* = 9 pairs; IPSC: sub = 7 ± 2 pA, supra = 54 ± 22 pA, p = 0.027; *n* = 9 pairs; 3 animals) (*Figure 5G–I*). These findings suggest that rBLA, but not cBLA, influences PL L5 PT neurons via local connections across layers of the PFC.

While cBLA does not evoke polysynaptic responses in PL L5 neurons, it is possible that IL L5 PT neurons could make local connections that amplify cBLA inputs to IL L2 and IL L5 CA neurons. To test

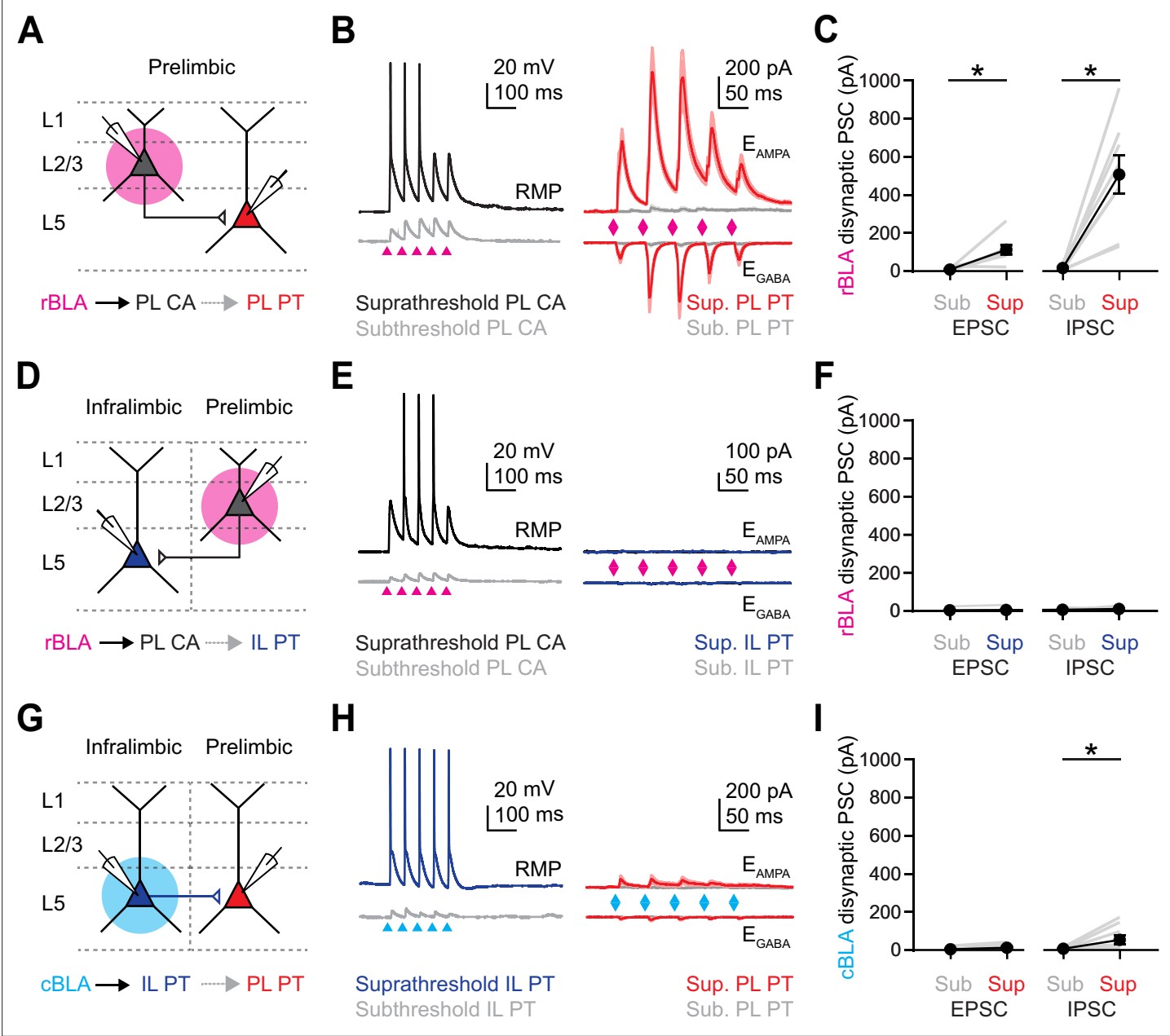

**Figure 5.** Rostral basolateral amygdala (rBLA) but not caudal basolateral amygdala (cBLA) engages prelimbic (PL) L5 pyramidal tract (PT) neurons via local networks. (**A**) Schematic for recording rBLA-evoked responses at PL L2 cortico-amygdalar (CA) neurons (black) and polysynaptic responses at PL L5 PT neurons (red). (**B**) *Left*, Example rBLA-evoked EPSPs and APs from PL L2 CA neurons recorded in current-clamp at resting membrane potential (RMP) with subthreshold (bottom, gray) and suprathreshold (top, black) stimulation. *Right*, Average rBLA-evoked excitatory postsynaptic currents (EPSCs) (bottom, at $E_{GABA}$) and IPSCs (top, at $E_{AMPA}$) recorded at PL L5 PT neurons at the same subthreshold (gray) and suprathreshold (red) stimulation. (**C**) Summary of rBLA-evoked $EPSC_1$ and $IPSC_1$ amplitudes at PL L5 PT neurons. Gray lines show subthreshold and suprathreshold responses in the same neuron ($n$ = 8 cells, 4 animals). (**D–F**) Similar to (**A–C**) for rBLA-evoked EPSPs and APs at PL L2 CA neurons (gray and black) and lack of polysynaptic responses at infralimbic (IL) L5 PT neurons (right, blue) ($n$ = 8 cells, 3 animals). (**G–I**) Similar to (**A–C**) for cBLA-evoked EPSPs and APs at IL L5 PT neurons (gray and blue) and lack of polysynaptic responses at PL L5 PT neurons (red) ($n$ = 9 cells, 3 animals). *$p < 0.05$. See also **Figure 5—figure supplement 1**.

The online version of this article includes the following figure supplement(s) for figure 5:

**Figure supplement 1.** Caudal basolateral amygdala (cBLA) does not appreciably engage the mPFC local circuit.

this idea, we also stimulated cBLA axons at 20 Hz while recording from triplets of IL L5 PT, IL L5 CA, and IL L2 CA neurons (*Figure 5—figure supplement 1*). However, even at stimulus intensities that drove robust suprathreshold responses in IL L5 PT neurons, we never observed any action potentials in the CA neurons, indicating that cBLA does not drive a reciprocal PFC to BLA projection.

## Layer 2 CA neurons target L5 but not L3 pyramidal neurons in PL PFC

Given that our in vivo and ex vivo data suggested rBLA evokes polysynaptic activity, we next examined whether L2 CA neurons mediate a descending translaminar connection that targets L5 PT neurons in PL. To selectively activated CA neurons in L2, we used the soma-targeted opsin st-ChroME (*Mardinly et al., 2018*). We injected AAVrg-Cre into the BLA, as well as AAV-DIO-st-ChroME into the PFC, allowing for expression in CA neurons (*Figure 6A*). In order to confirm selective activation, we recorded from either L2 or L5 CA neurons while stimulating with bars of LED stimulation parallel to the pia (*Figure 6B*). We found that L2 stimulation could exclusively drive L2 CA neurons without engaging L5 CA neurons, whereas L5 stimulation could drive L5 CA neurons without engaging L2 CA neurons, demonstrating restricted layer-specific activation (*Figure 6C, D*).

Using this approach, we next stimulated PL L2 CA neurons and recorded evoked EPSCs and IPSCs at triplets of pyramidal neurons across the layers (*Figure 6E*). We found prominent EPSCs at L2 and L5 pyramidal neurons, but minimal responses at L3 pyramidal neurons (EPSC: L2 Pyr = −61 ± 19 pA, L3 Pyr = −13 ± 3 pA, L5 Pyr = −237 ± 96 pA; L2 vs. L3, p = 0.0373; L3 vs. L5, p = 0.0014; n = 8 triplets; 4 animals) (*Figure 6F*). We also observed prominent inhibition at L2 and L5 pyramidal neurons, again with minimal responses at L3 pyramidal neurons (IPSC: L2 Pyr = 888 ± 263 pA, L3 Pyr = 80 ± 56 pA, L5 Pyr = 449 ± 236 pA; L2 vs. L3, p = 0.0014; n = 8 triplets; 4 animals) (*Figure 6F*). The overall effect was a significantly higher *E/I* ratio at L5 pyramidal neurons compared to the other cell types (*E/I* ratio: L2 Pyr = 0.06, GSD = 1.8, L3 Pyr = 0.36, GSD = 5.5, L5 Pyr = 0.8, GSD = 2.8; L2 vs. L5, p = 0.0059; n = 7 L2, 5 L3, and 8 L5 cells; 4 animals). Similar responses were found using AAV-DIO-ChR2, which leads to expression of ChR2 in both L2 and L5 CA neurons (*Figure 6—figure supplement 1*). Together, these recordings suggest that PL L2 CA neurons can mediate polysynaptic responses to rBLA inputs, leading to robust excitation and inhibition in deeper layers of the PFC.

## Layer 2 CA neurons selectively engage L5 PT neurons in PL PFC

Although our results show that L2 CA neurons contact L5 pyramidal neurons, they do not establish which neurons are targeted. It is well known that both intralaminar and interlaminar connectivity depends on the postsynaptic cell type, but the cell type-specific targeting from L2 CA neurons remains unknown (*Anderson et al., 2010*; *Hirai et al., 2012*). To examine this connectivity, we next stimulated L2 CA neurons at 20 Hz and recorded evoked responses from pairs of labeled PL L5 IT (intratelencephalic) or PL L5 PT neurons (*Figure 7A*). We observed robust PL L2 CA-evoked EPSCs and IPSCs, which were strongly biased onto L5 PT neurons compared to neighboring L5 IT neurons (EPSC: PL L5 PT = −67 ± 15 pA, PL L5 IT = −4 ± 1 pA, p = 0.0005; n = 12 pairs; 7 animals; IPSC: PL L5 PT = 229 ± 92 pA, PL L5 IT = 20 ± 15 pA, p = 0.03; n = 10 pairs; 6 animals) (*Figure 7B*). Similar results were also found using AAV-DIO-ChR2 to non-selectively activate both L2 and L5 CA neurons and measure responses at L5 IT and PT neurons (*Figure 7—figure supplement 1*). These results indicate that activation of PL L2 CA neurons, which are the main cell type activated by rBLA input, in turn generate a combination of biased excitation and inhibition at PL L5 PT neurons.

Lastly, we used dynamic-clamp recordings to determine how PL L2 CA activation influence the firing properties of L5 pyramidal neurons (*Carter and Regehr, 2002*; *McGarry and Carter, 2016*; *Anastasiades et al., 2018*). We injected scaled versions of excitatory and inhibitory conductances measured in our voltage-clamp recordings, testing if how different combinations of inputs influence firing at L5 PT and L5 IT neurons (*Figure 7C*). We found that excitatory conductances alone evoked robust spiking in L5 PT neurons but not L5 IT neurons, whereas injecting excitatory and inhibitory conductances resulted in significantly attenuated spiking of PT neurons (spikes per stimulus: PT *E* = 6.82 ± 0.54; PT *E* + *I* = 1.07 ± 0.49; IT *E* = 0 ± 0; IT *E* + *I* = 0 ± 0; n = 7 for each cell type, 4 animals) (*Figure 7D*). These results indicate that activation of PL L2 CA neurons drives PL L5 PT spiking, but this can be strongly regulated by local inhibitory circuits. Together, these results describe the multiple divergent pathways through which rBLA and cBLA influence activity in the PFC, indicating that different subregions of the BLA regulate multiple PFC outputs via both mono- and polysynaptic pathways (*Figure 7E*).

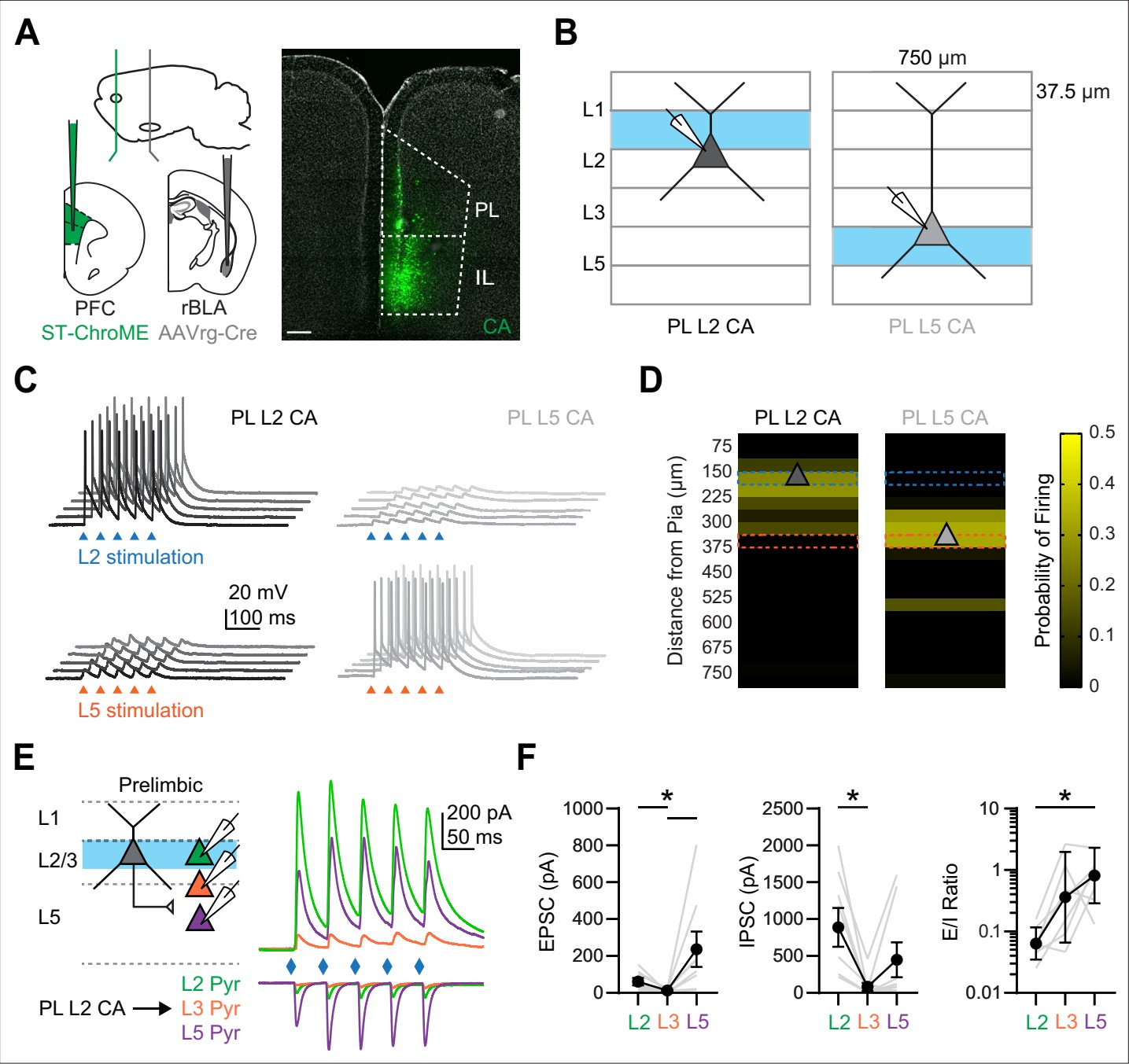

**Figure 6.** Prelimbic (PL) L2 cortico-amygdalar (CA) neurons evoke prominent responses in PL L5 pyramidal neurons. (**A**) *Left*, Schematic for injections of AAVrg-Cre into rostral basolateral amygdala (rBLA) and AAV-DIO-St-ChroME into prefrontal cortex (PFC). *Right*, Example image of CA neurons retrogradely labeled by AAVrg-Cre and expressing DIO-ChR2-eYFP (green) in the PFC. DAPI staining is shown in gray. Scale bar = 200 μm. (**B**) Schematic for laminar stimulation experiments, where st-ChroME + CA neurons in L2 and L5 of PL were recorded in current-clamp at resting membrane potential (RMP) and stimulated with 37.5 × 750 μm bars of light at 0.2 Hz. (**C**) Example traces from stimulation of L2 (top, blue) or L5 (bottom, orange) and evoked responses at PL L2 CA neurons (left, black) and PL L5 CA neurons (right, gray). Triangles denote light pulses. (**D**) Summary graphs of laminar stimulation control experiments at for PL L2 CA neurons (left) and PL L5 CA neurons (right) (*n* = 10 PL L2 CA neurons and 7 PL L5 CA neurons, 3 animals). Light-evoked firing is plotted as a function of the distance from the pia for a given laminar location. Triangle denotes average cell body location. (**E**) *Left*, Schematic for recording PL L2 CA-evoked responses at triplets of unlabeled L2 (green), L3 (orange), and L5 (purple) pyramidal neurons in PL. *Right*, Average PL L2 CA-evoked excitatory postsynaptic currents (EPSCs) (bottom, at $E_{GABA}$) and IPSCs (top, at $E_{AMPA}$) at the three pyramidal cell types. (**F**) *Left*, Summary of PL L2 CA-evoked EPSC$_1$ amplitudes for L2, L3, and L5 pyramidal neurons in PL, where gray lines denote individual triplets. *Middle*, Similar for IPSC$_1$ amplitudes. *Right*, Similar for EPSC$_1$/IPSC$_1$ ratio (*n* = 8 triplets, 4 animals). *p < 0.05. See also ***Figure 6—figure supplement 1***.

*Figure 6 continued on next page*

*Figure 6 continued*

The online version of this article includes the following figure supplement(s) for figure 6:

**Figure supplement 1.** Prelimbic (PL) cortico-amygdalar (CA) neurons project within the local network to L5.

## Discussion

We have determined how the rBLA and cBLA engage different subregions, layers, and cell types in the PFC. Anatomically, we found that non-overlapping neurons in the rBLA and cBLA send distinct axonal projections to the PL and IL. In slices, connections are cell type specific, with rBLA primarily contacting and driving PL L2 CA neurons, and cBLA mainly engaging IL L5 PT neurons. In the intact brain, cBLA activates IL L5, whereas rBLA both activates PL L2 and generates a mix of excitation and inhibition across L5. In the local circuit, PL L2 CA neurons mediate polysynaptic responses at PL L5 PT neurons, allowing rBLA inputs to influence subcortical output. Together, our findings reveal several new levels of organization for parallel circuits linking the BLA and PFC.

The BLA is integral for emotional behaviors, with function varying across the rostro-caudal axis (*Senn et al., 2014*; *Kim et al., 2016*; *Beyeler et al., 2018*). We focused on the rostral and caudal poles of the BLA, which correspond to the anatomically segregated and functionally opposed anterior BLA and posterior BLA (*Kim et al., 2016*; *Kim et al., 2017*; *Pi et al., 2020*). We found, rBLA projects to PL L2 and cBLA projects to IL L5, as had been described with the Rspo2-Cre and Cartpt-Cre transgenic lines (*Kim et al., 2016*). However, we also found that rBLA projects to IL L5, which was not seen in Rspo2-cre lines, and also evoked in vivo responses. This separate rBLA to IL L5 projection was also seen in a recent study of amygdalocortical connectivity, which defined multiple distinct projections from anterior BLA to PFC (*Hintiryan et al., 2021*). Equivalent connectivity is also found in primates, with magnocellular BLA projecting to superficial layers of area 32, a primate analogue of rBLA projecting to PL, and parvocellular BLA projecting more diffusely to L5 of area 25, a primate analogue of cBLA projecting to IL (*Sharma et al., 2020*). This homologous circuit organization suggests that divergent rBLA and cBLA projections to the PFC are a conserved element of amygdalocortical connectivity. There is likely additional diversity in these circuits, as gradients in cell types and projection targets are also observed in the medial-lateral and ventral–dorsal axes of the BLA (*McGarry and Carter, 2017*; *Beyeler et al., 2018*; *O'Leary et al., 2020*).

In the intact brain, BLA activity can evoke both excitation and inhibition of PFC neurons in a valence-specific manner (*Burgos-Robles et al., 2017*). In our in vivo recordings, we found that stimulation of rBLA and cBLA also causes distinct patterns of activation and suppression across subregions and layers of the PFC. rBLA exerts broad influence over most of PFC, with strong activation of PL L2, but also suppression of PL L5 and IL L5. While we expected a delay between PL L2 and PL L5 given our ex vivo physiology, we did not see any significant differences in the latency to fire among activated units, although technical limitations such as being unable to record L2 and L5 in a single recording, combined with relatively few activated units in PL L5, could obscure more subtle differences in spike timing. In contrast, cBLA generates a mix of activation and suppression in IL L5, but has only minimal influence on either PL L2 or PL L5. This evoked activity does not reflect a simple gradient of rBLA to dorsal PFC and cBLA to ventral PFC, with the divergence of L5 responses highlighting how rBLA and cBLA are not merely parallel pathways from a single input. Interestingly, IL L5 is particularly important for threat processing (*Adhikari et al., 2015*; *Bukalo et al., 2015*; *Bloodgood et al., 2018*), and we find it can be suppressed by rBLA but activated by cBLA inputs. In the future, it will be particularly important to determine how rBLA and cBLA projections to the PFC encode emotional valence, including threat processing.

Our slice physiology revealed cell type-specific connectivity, with rBLA primarily targeting reciprocally projecting PL L2 CA neurons, and cBLA almost exclusively targeting subcortically projecting IL L5 PT neurons that directly influence subcortical nuclei involved in threat responses (*Do-Monte et al., 2015*; *Bloodgood et al., 2018*; *Vander Weele et al., 2018*; *Huang et al., 2019*). These findings help reconcile previous studies, which showed how non-spatially restricted BLA inputs target PL L2 CA neurons (*Little and Carter, 2013*) and IL L5 PT neurons (*Cheriyan et al., 2016*). While IL L2 CA neurons also receive input from both rBLA and cBLA, these connections are weaker than onto the primary targets of each input. Interestingly, PL L5 PT neurons receive no direct input from either rBLA or cBLA, even though they are strongly implicated in mediating threat detection (*Rozeske et al.,*

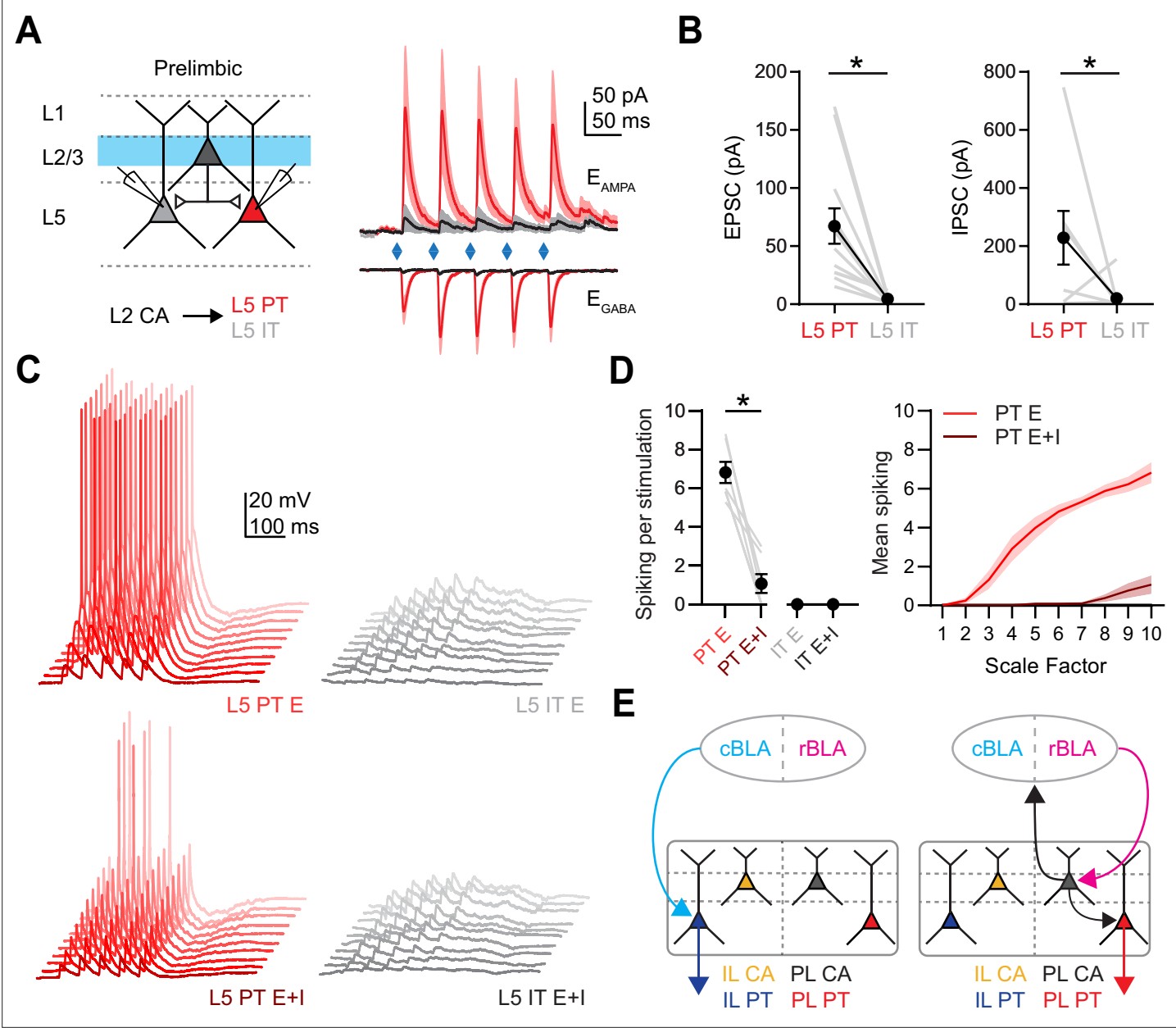

**Figure 7.** Prelimbic (PL) L2 cortico-amygdalar (CA) neurons specifically target PL L5 pyramidal tract (PT) over PL L5 IT neurons. (**A**) *Left*, Schematic for recording PL L2 CA-evoked responses at pairs of PL L5 IT and PL L5 PT neurons. *Right*, Average PL L2 CA-evoked excitatory postsynaptic currents (EPSCs) and IPSCs recorded at PL L5 IT (gray) and PL L5 PT (red) neurons. (**B**) *Left*, Summary of PL L2 CA-evoked $EPSC_1$ amplitudes at PL L5 IT and PL L5 PT neurons, where gray lines denote individual pairs. *Right*, Similar for $IPSC_1$ amplitudes ($n$ = 12 pairs for EPSCs and 10 pairs for IPSCs, 7 animals). (**C**) Example traces from dynamic-clamp recordings, where excitatory (*E*) or mixed excitatory and inhibitory (*E* + *I*) conductances derived from (**A**) were scaled and injected into respective PL L5 PT (red) or PL L5 IT (gray) neurons. Lighter shades represent higher scale factors. (**D**) *Left*, Summary average spiking elicited from injecting *E* or *E* + *I* conductances into respective cell types at maximum scaled conductance values. Gray lines denote spiking from individual neurons. *Right*, Summary of spiking evoked at all scale factors ($n$ = 7 for each cell type, 4 animals). (**E**) Revised models of synaptic connectivity for caudal basolateral amygdala (cBLA) to infralimbic (IL) (left) and rostral basolateral amygdala (rBLA) to PL (right). *p < 0.05. See also ***Figure 7—figure supplement 1***.

The online version of this article includes the following figure supplement(s) for figure 7:

**Figure supplement 1.** Prelimbic (PL) cortico-amygdalar (CA) neurons project preferentially to L5 pyramidal tract (PT) neurons.

*2018*; *Huang et al., 2019*). Instead, we found PL L5 PT neurons receive polysynaptic excitation from rBLA via PL L2 CA neurons, as well as inhibition via the local circuit. These findings provide a mechanism for how differences in cell type-specific targeting and activation from BLA to PL and IL arise from different presynaptic BLA subregions.

Several of our experiments indicate that rBLA and cBLA inputs evoke distinct patterns of excitation and inhibition across subregions and layers of the PFC. Our slice recordings show rBLA evokes inhibition in multiple cell types, including PL L2 CA, PL L5 PT, and IL L5 PT neurons. However, the *E/I* ratio was higher for PL L2 CA neurons, consistent with our in vivo studies showing this subregion and layer is primarily excited by these inputs. In contrast, our slice recordings show cBLA evokes prominent excitation and inhibition only at IL L5 PT neurons, again consistent with our in vivo studies. In the future, it will be important to establish which interneurons mediate rBLA and cBLA-evoked inhibition in L5 across subregions of the PFC, and determine if they are similar to interneurons targeted by glutamatergic inputs from contralateral cortex (*Anastasiades et al., 2018*), thalamus (*Anastasiades et al., 2021*), or hippocampus (*Liu et al., 2020*).

Repetitive rBLA and cBLA inputs generate cell type-specific firing in the PFC, allowing them to activate distinct output pathways either directly or via the local circuit. rBLA drives PL L2 CA neurons, consistent with previous work showing BLA engages in a reciprocal loop with PFC (*Little and Carter, 2013*; *McGarry and Carter, 2017*), which in turn evoke polysynaptic excitation and inhibition in PL L5 PT neurons. In contrast, cBLA strongly activates IL L5 PT neurons, has minimal influence on PL L2 CA, IL L2 CA, or IL L5 CA neurons, and evokes minimal polysynaptic responses in PL L5. Interestingly, evoked activity is different from ventral hippocampal inputs, which are another major afferent to IL and primarily engage L5 IT neurons (*Liu and Carter, 2018*). Our finding of minimal inter-subregion connectivity contrasts with prior work showing connections from IL L5/6 to PL L5/6 (*Marek et al., 2018*). One explanation is that communication is mediated by L5 IT neurons, which are poorly activated by cBLA inputs. In contrast, L5 PT neurons are a major output pathway across the cortex (*Economo et al., 2018*), and have relatively few connections locally (*Morishima and Kawaguchi, 2006*), consistent with the limited impact of cBLA activity on PL.

As a mechanism for how signals spread in the PFC, we found that PL L2 CA neurons project to PL L5 PT neurons and not neighboring PL L5 IT neurons. Interestingly, these connections bypass L3, reminiscent of superficial neurons crossing L4 when projecting to deeper layers of sensory cortex (*Lefort et al., 2009*; *Hooks et al., 2011*). The extreme bias in targeting of L5 PT over L5 IT also agrees with targeted connectivity from superficial to deep layers observed in other parts of frontal cortex (*Anderson et al., 2010*). It also suggests a fundamental difference in L2/3 connectivity within the local circuit between sensory and frontal cortices (*Hirai et al., 2012*; *Joshi et al., 2015*). Interestingly, our results are complemented by recent high-throughput circuit mapping of CA neuron local connectivity in PFC, which finds that CA neurons preferentially synapse onto other CA neurons, and that L2 and L5 CA neurons display distinct connectivity (*Printz et al., 2021*). Together, these findings indicate that PL L2 CA neurons serve as a crucial node in a disynaptic feedforward circuit, which links rBLA input specifically to PL L5 PT neurons without significant input to L3 or other L5 neurons. Consequently, the influence of rBLA on PFC output will depend on the neuromodulatory state, which may alter the tightly balanced CA-evoked excitation and inhibition onto PL L5 PT neurons (*Floresco and Tse, 2007*; *Vander Weele et al., 2018*; *Anastasiades et al., 2019*).

Long-range circuits involving the PFC exist in multiple motifs, including unidirectional connections (*Liu and Carter, 2018*), 'closed' reciprocal loops (*Little and Carter, 2013*), and 'open' reciprocal loops (*Collins et al., 2018*). Our results confirm that rBLA participates in a rare example of a closed reciprocal loop with PL, contacting L2 CA neurons that in turn project back to the rBLA (*Little and Carter, 2013*). These connections are also under inhibitory control via local interneurons, which prevent recurrent excitation (*McGarry and Carter, 2016*). In contrast, cBLA does not participate in a closed or open reciprocal loop with IL, instead making unidirectional connections onto L5 PT neurons. These connections differ from vHPC inputs, which are also unidirectional but instead contact IL L5 IT neurons (*Liu and Carter, 2018*), and also differ from thalamic inputs that instead contact L2/3 pyramidal neurons (*Collins et al., 2018*) and the apical dendrites of L5 PT neurons (*Anastasiades et al., 2021*). Instead, cBLA inputs appear most similar to callosal inputs from the contralateral PFC, which span multiple layers, but within L5 are also biased onto PT neurons, although not to the same degree (*Anastasiades et al., 2018*; *Anastasiades and Carter, 2021*).

In conclusion, our results determine how rBLA and cBLA make distinct projections to PL and IL, driving the activity of different output pathways via two unique circuits. Along with recent studies (*Kim et al., 2016*; *Zhang et al., 2021*), our results contribute to a new framework to study the role of BLA-PFC circuits, with important implications for understanding the role of this circuit in behavior. For example, BLA to PL projections are necessary for expression of threat conditioning (*Senn et al., 2014*; *Burgos-Robles et al., 2017*), and because rBLA is the main input to PL, our findings are consistent with the rBLA playing a key role in aversion (*Kim et al., 2016*; *Pi et al., 2020*). In contrast, BLA to IL projections are involved in the extinction of threat conditioning (*Senn et al., 2014*; *Klavir et al., 2017*), and because cBLA is the primary input to IL, our findings are also consistent with cBLA being involved in this learning (*Zhang et al., 2020*). What remains less clear is how these different streams of positive and negative valence may in turn bias PL toward encoding negative valence and IL toward encoding positive valence. Previous work suggests that PL and IL are functionally opposed on retrieval versus extinction of conditioned responses, respectively, regardless of the valence (*Sierra-Mercado et al., 2011*; *Burgos-Robles et al., 2017*; *Otis et al., 2017*; *Bloodgood et al., 2018*; *Cameron et al., 2019*). Further complicating this model of BLA function is recent work suggesting that there are neurons that encode both positive and negative valence within the rBLA (*Zhang et al., 2021*), which means both PL and IL could receive signals encoding positive and negative valence from different sources in the BLA. In the future, it will be important to explicitly test these ideas by measuring activity in specific subcircuits of the BLA and PFC during different forms of motivated behavior.

## Materials and methods

Experiments involved P28–P70 wild-type mice on a C57 BL/6J background. Experiments used male and female mice, and no significant differences were found between groups. All procedures followed guidelines approved by the New York University Animal Welfare Committee (protocols 07-1281 and 18-1503).

## Stereotaxic injections

P28–P56 mice were deeply anesthetized with either isoflurane or a mixture of ketamine and xylazine and head fixed in a stereotax (Kopf Instruments). A small craniotomy was made over the injection site, through which retrograde tracers and/or viruses were injected using a Nanoject III (Drummond). Injection site coordinates were relative to bregma (mediolateral, dorsoventral, and rostro-caudal axes: PL PFC = ±0.35, −2.1, +2.2 mm; IL PFC = ±0.35, −2.3, +2.2 mm; rBLA = −3.0, −5.1, −1.1 mm; cBLA = −3.0, −5.1, −1.7 mm; PAG = −0.6, −3.0 and −2.5, −4.0 mm). Dual AAVrg experiments were injected at +2.4 mm for PL and +2.0 mm for IL in the rostro-caudal axis to minimize viral leak. Borosilicate pipettes with 5–10 µm tip diameters were back filled, and 100–500 nl was pressure injected, with 30–45 second inter-injection intervals.

For retrograde labeling or confirmation of injection site for electrophysiology experiments, pipettes were filled with either CTB conjugated to Alexa 488, 555, or 647 (Life Technologies) or blue latex beads. AAV1-hSyn-hChR2-eYFP (UPenn Vector Core AV-1-26973P/Addgene 26973-AAV1) or AAV1-CamKIIa-hChR2-mCherry (UPenn Vector Core AV-1-26975/Addgene 26975-AAV1) were used for non-conditional axon labeling. AAVrg-CAG-tdTomato (Addgene 59462-AAVrg), AAVrg-CAG-GFP (Addgene 37825-AAVrg), and AAVrg-mCherry-IRES-Cre (Addgene 55632-AAVrg) were used for retrograde labeling in histological experiments. Optogenetic stimulation was achieved using AAV1-hSyn-hChR2-eYFP, AAV1-EF1a-DIO-hChR2-eYFP (UPenn Vector Core AV-1-20298P/Addgene 20298-AAV1) or AAV9-CAG-DIO-ChroME-ST-p2A-H2B-mRuby (generously provided by Hillel Adesnik). The St-ChroME virus was diluted 1:10 in 0.01 M phosphate-buffered saline (PBS) prior to injection. Simultaneous virus and tracer injections were mixed in a 1:1 virus:tracer ratio, except for experiments involving axonal stimulation evoked firing, which used a 3:2 virus:tracer ratio. Following injections, the pipette was left in place for an additional 10 min before being slowly withdrawn to ensure injections remained local. Retrograde-Cre experiments to label CA neurons were carried out in a similar manner, with injection of AAVrg-hSyn-Cre (Addgene 105553-AAVrg). After all injections, animals were returned to their home cages for 2–4 weeks before being used for experiments, except for experiments involving st-ChroME virus, which were returned for 7–9 days to minimize expression time.

## In vivo electrophysiology

Two to four weeks after viral injections into either rBLA or cBLA, mice were anesthetized with isoflurane, the skin overlying the skull was surgically removed, and a custom plate was attached to the skull using Meta-bond (Parkell), leaving the area over the PFC exposed. An AgCl reference electrode was implanted contralateral to the recording site, attached to an external gold pin. A multimode, 200 µm ID fiber optic connected to a 1.25-mm steel ferrule (Thorlabs) was implanted in either the rBLA or cBLA. The exposed skull was then covered with Kwik-Cast (WPI), and mice were allowed to recover in the home cage for at least 3 days before recording.

On the day of recording, mice were anesthetized with isoflurane and a small craniotomy was made over the PFC, using the stereotaxic coordinates designated above. After recovering from anesthesia, mice were then head-fixed using the implanted plate and allowed to run freely on a spinning treadmill. A 473-nm LED was attached to the ferrule with a 200-µm ID patch cable (Thorlabs), allowing stimulation of rBLA or cBLA. Before insertion into the brain, an NP electrode array (*Jun et al., 2017*) was mounted on a micromanipulator (Sutter Instruments) and painted with DiI (Thermo Fisher, 2 mg/ml in ethanol) for post hoc track reconstruction. The NP probe was lowered vertically 3 mm below the dorsal brain surface and, laterally from the midline, either 350–450 µm for L5 recordings or 150–250 µm for L2/3 recordings. The probe was advanced slowly (~2 µm/s) and allowed to rest for 30 min before recording. A drop of silicone oil (1000cs, Dow Corning) was placed over the exposed brain to prevent drying. Probes were referenced to the implanted Ag/AgCl wire. Analog traces were filtered (0.3–5 kHz), digitized, and recorded (30 kHz per channel) using acquisition boards from National Instruments and OpenEphys software (*Jun et al., 2017*). NP recordings were made in external reference mode with LFP gain of 250 and AP gain of 500. Either rBLA or cBLA was photostimulated at 20 Hz for 5 pulses of 2-ms duration, repeated for 40 trials with an intertrial interval of 30 s. Following recordings, brains were collected and processed for post hoc probe track reconstruction.

## Slice preparation

Mice were anesthetized with an intraperitoneal injection of a lethal dose of ketamine and xylazine and then perfused intracardially with an ice-cold cutting solution containing the following (in mM): 65 sucrose, 76 NaCl, 25 NaHCO$_3$, 1.4 NaH$_2$PO$_4$, 25 glucose, 2.5 KCl, 7 MgCl$_2$, 0.4 Na-ascorbate, and 2 Na-pyruvate (bubbled with 95% O$_2$/5% CO$_2$). 300 µm coronal sections were cut in this solution and transferred to ACSF containing the following (in mM): 120 NaCl, 25 NaHCO$_3$, 1.4 NaH$_2$PO$_4$, 21 glucose, 2.5 KCl, 2 CaCl$_2$, 1 MgCl$_2$, 0.4 Na-ascorbate, and 2 Na-pyruvate (bubbled with 95% O$_2$/5% CO$_2$). Slices were recovered for 30 min at 35°C and stored for at least 30 min at 24°C. All experiments were conducted at 30–32°C.

## Slice electrophysiology

Targeted whole-cell recordings were made from projection neurons in PL and IL using infrared-differential interference contrast. In the PFC, layers were defined by distance from the pial surface: L2: 170–220 µm; L3: 225–300 µm; L5: 350–550 µm; L6: 600–850 µm. PL and IL were defined as 800–1200 and 1600–2000 µm from the dorsal surface, and the intermediate ventral PL area was deliberately avoided. CA, CC, IT, and PT neurons were identified by the presence of fluorescently tagged CTB. For recordings from st-ChroME + CA neurons, neurons in L2 or L5 were identified by mRuby expression.

For voltage-clamp experiments, borosilicate pipettes (3–5 MΩ) were filled with (in mM): 135 Cs-gluconate, 10 4-(2-hydroxyethyl)-1-piperazineethanesulfonic acid (HEPES), 10 Na-phosphocreatine, 4 Mg$_2$-ATP, 0.4 NaGTP, 10 CsCl, and 10 ethylene glycol-bis(β-aminoethyl ether)-N,N,N′,N′-tetraacetic acid (EGTA), pH 7.3 with CsOH (290–295 mOsm). For current- and dynamic-clamp recordings, borosilicate pipettes (3–5 MΩ) were filled with (in mM): 135 K-gluconate, 7 KCl, 10 HEPES, 10 Na-phosphocreatine, 4 Mg$_2$-ATP, 0.4 NaGTP, and 0.5 EGTA, pH 7.3 with KOH (290–295 mOsm). In some voltage-clamp experiments, 1 µM tetrodotoxin (TTX) was included in the bath to block action potentials (APs), along with 0.1 mM 4-aminopyridine (4-AP) and 4 mM external Ca$^{2+}$ to restore presynaptic glutamate release. In some current-clamp experiments, 10 µM 3-(2-Carboxypiperazin-4-yl)propyl-1-phosphonic acid (CPP) and 10 µM gabazine were included to block NMDA receptors and GABA$_A$ receptors, respectively. In voltage-clamp experiments recording IPSCs, 10 µM CPP was included to block NMDA receptors. In dynamic-clamp recordings, 10 µM 2,3-dihydroxy-6-nitro-7-sulphamoyl-ben

zo(F)quinoxaline (NBQX), 10 µM CPP, and 10 µM gabazine were included to block AMPA, NMDA, and GABA$_A$ receptors, respectively. All chemicals were from Sigma or Tocris Bioscience.

Electrophysiology data for voltage- and current-clamp experiments were collected with a Multi-clamp 700B amplifier (Axon Instruments) and National Instruments boards using custom software in MATLAB (MathWorks). Dynamic-clamp recordings were performed using an ITC-18 interface (Heka Electronics) running at 50 kHz with Igor Pro software (Wavemetrics) running MafPC (courtesy of Matthew Xu-Friedman). For dynamic-clamp recordings, conductances were initially converted from st-ChroME local-circuit mapping experiments and then injected into neurons while being multiplied by a range of constant scale factors to model the effects of driving the L2 CA input at various stimulus strengths. The reversal potentials for AMPA-R excitation and GABAa-R inhibition were set to 0 and −75 mV, respectively. Signals were sampled at 10 kHz and filtered at either 5 kHz for current- and dynamic-clamp recordings, or 2 kHz for voltage-clamp recordings. Series resistance was measured as 10–25 MΩ and not compensated.

## Slice optogenetics

Glutamate release was triggered by activating channelrhodopsin-2 (ChR2) present in the presynaptic terminals of rBLA or cBLA inputs to the PFC (*Little and Carter, 2012*; *McGarry and Carter, 2016*). ChR2 was activated with 2ms pulses of 473 nm light from a blue LED (473 nm; Thorlabs) through a 10 × 0.3 NA objective (Olympus) with a power range of 0.4–12 mW. LED power was adjusted until responses >100 pA were seen in at least one neuron in a recorded pair or triplet, with the same power used for all neurons in the slice. In some experiments, ChR2 was activated by 20-Hz stimulation with 2-ms pulses of light. Subcellular targeting recordings utilized a digital mirror device (Mightex Polygon 400 G) to stimulate a 10 × 10 grid of 75 µm squares at a power range of 0.05–0.2 mW per square at 1 Hz, with the first row aligned to the pia. For all other recordings, the objective was centered over the soma, unless noted otherwise. Intertrial interval was 10 s except for experiments involving trains of stimulation, in which it was 30 s.

## Soma-restricted optogenetics

To map the outputs of st-ChroME + CA neurons, stimulation parameters were first developed to produced robust, spatially restricted AP firing. Recordings were made from PL L2 and L5 st-ChroME + CA neurons located in the same slice of PFC. Blue (473 nm) LED light was illuminated as 37.5 × 750 µm bars using a DMD through a 10 × 0.3 NA objective. Individual bars were stimulated with 5 pulses of 1 ms light at 20 Hz with an intensity of 0.1–0.8 mW in a pseudorandom order. Responses were recorded at RMPs. High power (0.8 mW) stimulation resulted in st-ChroME + neurons firing outside of their layer and therefore was not used in subsequent experiments. In experiments involving the postsynaptic targeting of CA neurons onto CC and PT neurons, the fourth or ninth bars from the pia, which are equivalent to L2/3 or L5, were alternatingly stimulated at 0.1–0.4 mW, with a 15-s interstimulus interval.

## Histology

Mice were anesthetized and perfused intracardially with 0.01 M PBS followed by 4% paraformaldehyde (PFA). Brains were stored in 4% PFA for 12–18 hr at 4°C before being washed three times (30 min each) in 0.01 M PBS. Slices were cut on a VT-1000S vibratome (Leica) at 100 µm thickness, except for NP tract reconstruction, which were cut at 70 µm thickness, and then placed on gel-coated glass slides. ProLong Gold anti-fade reagent with DAPI (Invitrogen) or VectaShield with DAPI (Vector Labs) was applied to the surface of the slices, which were then covered with a glass coverslip. Fluorescent images were taken on an Olympus VS120 microscope, using a 10 × 0.25 NA objective (Olympus). For rBLA and cBLA axonal projections to the PFC and retrograde cell counting in the BLA, images were taken on a TCS SP8 confocal microscope (Leica), using a 10 × 0.4 NA objective or 20 × 0.75 NA objective (Leica).

## Data analysis

Slice electrophysiology was analyzed using Igor Pro (WaveMetrics), except for MATLAB (MathWorks) for st-ChroME control experiments and sCrACM recordings. For voltage-clamp recordings, PSC amplitudes were measured as the average at 1ms around the peak response.

In vivo electrophysiology data were preprocessed by referencing to the common median across all channels. The data were then spike-sorted using Kilosort2 (*Pachitariu et al., 2016*; *Pachitariu et al., 2020*), with manual curation in Phy (*Rossant et al., 2016*; *Rossant et al., 2020*). Spike time and waveform data were further processed and visualized using MATLAB code modified from N. Steinmetz (*Steinmetz et al., 2021*). Importantly, all units were then aligned to brain location using the Allen CCF (*Wang et al., 2020*). Significantly responsive single units were determined by using a Wilcoxon signed-rank test (threshold of $p < 0.05$), comparing the number of spikes in a 2-s baseline period before LED stimulation and the number of spikes 5–25 ms after every LED pulse. If the unit had significantly lower firing during the LED window and an average $z$-scored value of $-0.2$ or less, it was designated as suppressed. If the unit had significantly higher firing during the LED and an average $z$-scored value of $+0.2$ or more, it was designated as significantly activated. Chi-squared tests were used to compare the proportion of significantly activated or inhibited units across PFC in response to rBLA or cBLA stimulation and were corrected for multiple comparisons by the Holm–Bonferroni method.

Summary data are reported in the text and figures as arithmetic mean ± standard error of the mean (SEM) or geometric mean and GSD factor for ratio data. In some graphs with three or more traces, SEM waves are omitted for clarity. *E/I* ratios were only computed for neurons with an average $IPSC_1$ greater than 5 pA. Statistical analyses were performed using Prism 9 (GraphPad Software). Comparisons between unpaired data were performed using two-tailed Mann–Whitney tests. Comparisons between data recorded in pairs were performed using two-tailed Wilcoxon matched-pairs signed rank tests. Ratio data were log-transformed and compared to a theoretical median of 0. For paired comparisons of more than two groups, Friedman tests with Dunn's multiple comparisons tests were performed. For unpaired comparisons of more than two groups, Kruskal–Wallis tests with Dunn's multiple comparisons tests were performed. For all tests, significance was defined as $p < 0.05$.

Cell counting in the BLA was performed using ImageJ on a multicolor image of retrogradely labeled neurons. ROIs for rBLA and cBLA were calculated in each slice after aligning to the Allen Brain Atlas at the appropriate rostro-caudal coordinate. The number of cells per slice was averaged across three animals and used to calculate averages ± SEM across animals. Axon distributions in the PFC were quantified using unbinned fluorescence profiles relative to distance from the pia for three slices from each animal. The subtract background function in FIJI was used followed by peak normalizing to a value of 100 and rescaling the minimum fluorescence to 0 on a per-slice basis for each fluorophore in our dual anterograde tracing experiments (*Figure 1D–F*) and our cre-dependent single fluorophore tracing experiments (*Figure 1—figure supplement 2*).

## Acknowledgements

We thank the Carter lab, Christine Constantinople, and Robert Froemke for helpful discussions and comments on the manuscript. This work was supported by NIH T32 MH019524 (KM) and NIH R01 MH085974 (AGC). The authors have no financial conflicts of interest.

## Additional information

### Funding

| Funder | Grant reference number | Author |
| --- | --- | --- |
| National Institute of Mental Health | R01 MH085974 | Adam G Carter |
| National Institute of Mental Health | T32 23 MH019524 | Kasra Manoocheri |

The funders had no role in study design, data collection, and interpretation, or the decision to submit the work for publication.

## Author contributions
Kasra Manoocheri, Conceptualization, Data curation, Formal analysis, Investigation, Visualization, Writing - original draft, Writing - review and editing; Adam G Carter, Conceptualization, Supervision, Funding acquisition, Writing - original draft, Project administration, Writing - review and editing

## Author ORCIDs
Kasra Manoocheri ⓘ http://orcid.org/0000-0001-9968-3478
Adam G Carter ⓘ http://orcid.org/0000-0003-2095-3901

## Ethics
All procedures followed guidelines approved by the New York University Animal Welfare Committee (protocols 07-1281 and 18-1503).

## Decision letter and Author response
Decision letter https://doi.org/10.7554/eLife.82688.sa1
Author response https://doi.org/10.7554/eLife.82688.sa2

---

## Additional files

### Supplementary files
- Transparent reporting form
- Source data 1. Statistical tests for all figures.
- Source data 2. Graph data for all figures.

### Data availability
We have included all statistical tests and results in Source data 1 and the data for figures in Source data 2.

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
