## [Editor Report]

This paper will be of interest to readers studying the neuronal circuit in general and those studying the basolateral amygdala (BLA), prefrontal cortex (PFC), and diseases associated with these regions. Using innovative circuit analysis techniques, this important paper shows that different subregions of the BLA form cell-type-specific connections with the subregion of PFC and engage specific local circuits within it. The key claim of the paper is supported by compelling data from multiple approaches.

---

## [Decision Letter]

**Decision letter after peer review:**

[Editors’ note: the authors submitted for reconsideration following the decision after peer review. What follows is the decision letter after the first round of review.]

Thank you for submitting the paper "Rostral and caudal BLA engage distinct circuits in the prelimbic and infralimbic PFC" for consideration at *eLife*. Your initial submission has been assessed by a Senior Editor in consultation with members of the Board of Reviewing Editors. Although the work is of interest, we are not convinced that the findings presented have the potential significance that we require for publication in *eLife*.

Specifically, the reviewers find that the work is expertly performed and provides careful characterization of physiologically demonstrated connections between BLA and mPFC. However, they note that this is an active area of research, with many recent publications on connectivity between these regions, albeit mostly with anatomical approaches so far, thus somewhat reducing the novelty of the findings. Further, the reviewers felt that the proposed parallel circuit motifs would need broader experimental support. If the authors are interested, we would be interested in seeing a substantially revised new paper that addresses the criticisms outlined in the detailed individual reviews below.

*Reviewer #1 (Recommendations for the authors):*

In this manuscript Manoocheri and Carter describe two parallel circuits that originate from two subregions in the basal-lateral amygdala, rBLA and cBLA, projecting to two subregions in the prefrontal cortex, PL and IL in a cell type-specific fashion. The strength of the current work includes: the importance of obtaining the detailed circuits in order to understand the amygdala-PFC function; the slice physiology experiments, which is the main readout of the work, is expertly done; the finding that local circuits in PL mediate some of BLA input is interesting. The enthusiasm for the current work is greatly dampened because a series of previous studies already confirmed the major hypothesis that the current work aimed to test and there is significant overlap in the details of the finding, reducing the novelty of the work. The central statement that 'it remains completely unknown how this (BLA influences on the PFC) differs between rBLA and cBLA inputs' is not really accurate. For example, Kim et al. 2016 showed that anterior/rostral BLA projects to Pl and posterior/caudal BLA projects to IL and these two circuits are segregated at gene expression level, anatomical level, and behavioral and functional level. The major conclusion of the current work that there are parallel rBLA-PL and cBLA-IL circuits seems like a recapitulation of Figure 7B in Kim et al. Follow up work in Zhang et al. 2020 further illustrated the function of PL projecting BLA neurons in fear extinction. In addition, recent anatomy studies of comprehensive BLA projections (Hintiryan et al., 2021) confirmed the notion of differential projections from BLA to PL and IL and provided further refinement of the subregion specificity in the anterior BLA projects to the PL and IL, some of which appeared to be different from the current study. E.g., Hintiryan et al. showed that rBLA project to both PL and IL and the origins differ within the anterior BLA. These potential differences are not addressed in the current work. Such published circuit mapping, together with previous in vitro slice physiology work (including beautiful papers from the Cater lab) that described BLA inputs to L2 in PL, L5 in IL, and inhibitory network in IL allows the deduction of the significant amount of conclusion here. Although some new details added, the relatively large body of previous work further reduces the impact of this work.

1. It will be important to show full injection sites (not just one section), particularly for targeting PL and IL since they are in close proximity. In fact, the example in Figure 1B looks like there is contamination in PL when targeting IL.

2. The projections shown in Figure1E suggest that axons of both colors have strong presence in the IL and PL, and in Figure1-supplent 1C axons of both colors have strong presence in PL. Is this different from the authors interpretation? These data actually are more consistent with the conclusions in Hintiryan et al. that PL actually receives both inputs from rBLA and cBLA. This might also explain the detected responses in IL when stimulating rBLA: there was significant number of neurons responded, although smaller population. Also, it might be useful to add quantification in Figure1-supplent 1C, similar to Figure 1F.

3. There are strong presence of IL CA neurons located in L5 (Figure 3B) but their responses were not mentioned. According to the example images, it seems like CA neurons are predominantly located in the IL and PT neurons are predominantly located in the PL. This bias looks comparable if not stronger than the axonal preference in Figure 1E. How would this impact Figure 6 diagram?

4. Are Figure 4 C and F essentially a reproduction of Figure 3C and D? If so, might consider move them to supplement figure.

*Reviewer #2 (Recommendations for the authors):*

This is a solid, well-designed study that provides a new level of anatomical and functional dissection of the BLA-mPFC circuit and is an important contribution to the field. The authors skillfully combined state-of-the-art techniques and the manuscript is well-written. My main criticisms are listed below and I would be happy to see the authors' response to these points:

• While the findings include a substantial amount of in vitro and in vivo electrophysiology, the potential behavioral roles of these distinct pathways remain mysterious. Is there any indication that stimuli with positive or negative valence differentially trigger activity in these mPFC regions, as would be expected from selective activation of the rBLA and cBLA in response to such stimuli? While such an experiment might be beyond the scope of the current manuscript, it would be good if the authors elaborate in the Discussion about the potential importance of this finding to value coding in the PFC.

• The retrograde tracing in Figure 1 was done by injecting one AAVretro more anterior to the other. While it's understandable that the authors wished to avoid leak between the two sites, it seems important to verify the results (more rBLA cells labeled with the more rostral AAVretro injection) using single viral injections, or a set of mPFC injections done at the same AP location.

• The use of patterned blue light, combined with expression of either ChR2 or st-Chrome, needs to be better validated. Can the authors show that by using st-Chrome they are eliminating any direct responses to stimulation of cells in other layers? How were the conditions (light bar size, intensity) chosen for this experiment? This is an important calibration experiment and should be included in the manuscript.

• The recordings done using ChR2 (Figure 6S1) seem redundant with the st-Chrome recordings in Figure 6. Given the constraints of using a non-soma-targeted opsin in this experiment (which the authors nicely explain in the text), it might be best to remove this data or acquire similar data with st-Chrome.

• The authors state (p.18) that "cBLA does not participate in a reciprocal loop with IL, instead making unidirectional connections onto L5 PT neurons"; however, it's clear from the images (for example. Figure 6A) that the IL contains a fair amount of CA cells as well. Given the previous work by the Carter lab on reciprocal BLA-mPFC circuits, it would have been very interesting to see how these L5 CA cells compare with the PT cells in their inputs from the cBLA and from IL L2 CA cells.

*Reviewer #3 (Recommendations for the authors):*

Emerging evidence suggests the BLA are anatomically segregated based on molecular markers, axonal projection, and function. This paper further strengthened this view by identifying the rostral and caudal BLA (rBLA and cBLA) as a distinct subregion based on the projection and connection to prelimbic (PL) and infralimbic (IL) part of the PFC. Specifically, the paper demonstrated projection from rBLA and cBLA are connected most strongly to the L2 cortico-amygdala (CA) neurons in PL and L5 PT neurons in IL, respectively, and are capable of evoking spikes preferentially in these neurons/layers both in slice and in vivo. They also found little activity spread between IL and PL upon stimulation of r- or cBLA axons in each PFC subregion. Moreover, L2 CA neurons in PL are preferentially connected to the L5 PT neurons than L5 IT neurons locally. The data support the paper's main conclusion and illuminate the mechanistic underpinning of functional distinction within the BLA and PFC reported in earlier studies.

The strength of the paper is the use of a wide variety of techniques to verify the subregion-specific connections between BLA and PFC and demonstrate this connectivity bias has functional relevance (i.e., causing subregion- and layer-specific activation of neurons with distinct local connectivity). No major weakness in technicality or data interpretation is noted.

1. The PL and IL (and ACC on this matter) are ambiguously defined. Please clarify how the authors distinguished them visually.

2. The retrograde tracer injection is targeted at L5 region of PL and IL (mediolateral 0.35 mm), despite in PL, it is not a major projection site for r- or cBLA. It will strengthen the paper if the degree of retrograde labeling bias in Figure 1C is shown to be preserved when tracer injection is targeted into the superficial layers, which appear to be the main projection site for both r- and cBLA (at least anatomically, as in Figure 1E).

3. In Figure 3B, L5 PT neurons (labeled by retrograde tracer injection into PAG) is mostly in PL (and at the border between PL and IL). This makes me wonder how the approach reliably labeled L5 PT neurons in IL, and whether the PAG-projecting PT neurons are the representative subtype of PT neurons in IL. If this is not a representative image then I suggest replacing it with one that is.

4. The latency of L5 PL spiking in vivo is similar between L2 CA and L5 PT neurons in PL and appears to be contrary to the polysynaptic circuit model (L2 CA to L5 PT) proposed to be engaged by rBLA. It would be helpful if the authors could discuss this discrepancy.

5. Color code for L2 CA IL and PL L5 PT is hard to distinguish (e.g., Figure 3C). I recommend using different color codes.

6. The vertical lines other than the Neuropixel tract are quite distracting in Figure 2. Supp 1. I suggest removing them if possible.

7. Line 371: Figure S5 should be Figure 5

[Editors’ note: further revisions were suggested prior to acceptance, as described below.]

Thank you for resubmitting your work entitled "Rostral and caudal BLA engage distinct circuits in the prelimbic and infralimbic PFC" for further consideration by *eLife*. Your revised article has been evaluated by John Huguenard (Senior Editor) and a Reviewing Editor.

The manuscript has been improved, but there are some remaining issues that need to be addressed, as outlined by reviewer 2 below:

Note that all reviewers felt that the main strength of this manuscript is in the detailed synaptic physiology, whereas the anatomical identification of the cells as CA/PT is fuzzier and might inherently lump together multiple different functional classes

*Reviewer #1 (Recommendations for the authors):*

The revised manuscript by Manoocheri and Carter addressed my concerns. Furthermore, in my opinion, the new logical flow made the manuscript much easier to follow.

*Reviewer #2 (Recommendations for the authors):*

Recommendation #1:

Please add more detail regarding the injection sites (Figure 1 and Figure 1- S1 show one representative section in each animal). With the newly added soma sections in 1B, it will be interesting to state whether there are double-labeled neurons (magenta and cyan).

Related, it will be good to explain in Figure1-S2 B experiment, whether the injection of AAVrg-cre mostly resides in L2 (Similar experiments in Figure 1-S2F nicely show that injection covers the entire IL). This will strengthen the observation that rBLA mainly projects to L2 in PL, and this is not due to that the injection only targets L2.

Recommendation #2:

Please add quantification in Figure 2A image for layer distribution which will help readers appreciate the cell distribution of the projection neurons, and also cite Ferreira et al., 2015 paper which has the same PAG and amygdala injection and quantified the distribution.

*Reviewer #3 (Recommendations for the authors):*

The revised manuscript is improved in clarity and does a better job placing the study's findings in the broader context of the field. I appreciated the addition of anatomical experiments to remove any doubt regarding the non-overlapping injections of AAVretro and the clarifications about the deep-layer cortico-amygdala neurons in the IL region.

---

## [Author Response]

[Editors’ note: the authors resubmitted a revised version of the paper for consideration. What follows is the authors’ response to the first round of review.]

Reviewer #1 (Recommendations for the authors):In this manuscript Manoocheri and Carter describe two parallel circuits that originate from two subregions in the basal-lateral amygdala, rBLA and cBLA, projecting to two subregions in the prefrontal cortex, PL and IL in a cell type-specific fashion. The strength of the current work includes: the importance of obtaining the detailed circuits in order to understand the amygdala-PFC function; the slice physiology experiments, which is the main readout of the work, is expertly done; the finding that local circuits in PL mediate some of BLA input is interesting. The enthusiasm for the current work is greatly dampened because a series of previous studies already confirmed the major hypothesis that the current work aimed to test and there is significant overlap in the details of the finding, reducing the novelty of the work. The central statement that 'it remains completely unknown how this (BLA influences on the PFC) differs between rBLA and cBLA inputs' is not really accurate. For example, Kim et al. 2016 showed that anterior/rostral BLA projects to Pl and posterior/caudal BLA projects to IL and these two circuits are segregated at gene expression level, anatomical level, and behavioral and functional level. The major conclusion of the current work that there are parallel rBLA-PL and cBLA-IL circuits seems like a recapitulation of Figure 7B in Kim et al. Follow up work in Zhang et al. 2020 further illustrated the function of PL projecting BLA neurons in fear extinction. In addition, recent anatomy studies of comprehensive BLA projections (Hintiryan et al., 2021) confirmed the notion of differential projections from BLA to PL and IL and provided further refinement of the subregion specificity in the anterior BLA projects to the PL and IL, some of which appeared to be different from the current study. E.g., Hintiryan et al. showed that rBLA project to both PL and IL and the origins differ within the anterior BLA. These potential differences are not addressed in the current work. Such published circuit mapping, together with previous in vitro slice physiology work (including beautiful papers from the Cater lab) that described BLA inputs to L2 in PL, L5 in IL, and inhibitory network in IL allows the deduction of the significant amount of conclusion here. Although some new details added, the relatively large body of previous work further reduces the impact of this work.

We appreciate the reviewer’s comments, but we disagree with their assertion that “previous studies already confirmed the major hypothesis” of our study. It is true that previous anatomy studies have shown distinct rBLAàPL and cBLAàIL projections. For example, the cited Kim et al. 2016 paper shows two populations of neurons within the amygdala, and their Figure 7B is similar to our Figure 1B/C, whereas their Figure 7O/P is similar to our Figure 1E/F, which is all anatomy. However, we note that the follow-up work of Zhang et al. 2020 is a behavior study, which is focused entirely within the cBLA, and has no overlap with any of our experiments. The recent Hintiryan et al. 2021 paper is an elegant and rigorous anatomy study of amygdala neurons and their connections, including in the PFC. However, we disagree that our results conflict, as we also see direct rBLA projections to both PL and IL. That said, we agree that paper reveals additional anatomical details that we did not explore, including a model for further segregation of rBLA, and we delve into how our findings compare in our revised Discussion.

In contrast to these papers, our own work primarily focuses on the physiological impact of different connections at different cell types. As noted by the reviewer, we and others have found that BLA input targets L2 CA neurons in PL and L5 PT cells in IL. The current paper reconciles these findings by showing that PL L2 CA neurons and IL L5 PT cells are targeted and activated by rBLA and cBLA, respectively (Figures 2 and 3). It also moves well beyond those studies to look at the polysynaptic networks that are engaged within the PFC, providing a mechanism for how BLA inputs can ultimately influence PT output from PL without direct connections (Figures 5-7). Lastly, it also advances beyond prior slice physiology to show how networks are differentially engaged as a result of rBLA or cBLA stimulation within the intact brain (Figure 4). We are unaware of any other work showing the in vivo activity that occurs when specifically driving rBLA or cBLA inputs to PL and IL. The foundational work in Kim et al. 2016 and Zhang et al. 2020 does not relate rBLA or cBLA activation with downstream activity in PFC, so there is relatively little known about the actual physiologic role of these circuits. Therefore, while we agree Figure 1 recapitulates anatomical findings and validates our ability to specifically target rBLA and cBLA with viral injections, Figures 2-7 contain an abundance of new experiments. Finally, we note that previous papers looking at these circuits have not disambiguated rBLA and cBLA inputs to the PFC. In many cases, they have also not distinguished between the different populations of postsynaptic neurons in the PFC. Our work shows that considering both inputs and cell types is critical for understanding how the BLA and PFC interact with each other.

In our revision, we more clearly describe the previous anatomical and behavioral studies that motivated our experiments. However, we hope the reviewer now appreciates that our results help to fill an important gap between anatomy and behavior. We think our findings are an important advance, which be of broad interest and have a genuine impact on the field.

1. It will be important to show full injection sites (not just one section), particularly for targeting PL and IL since they are in close proximity. In fact, the example in Figure 1B looks like there is contamination in PL when targeting IL.

The reviewer raises an important point about our anatomy, related to those raised by Reviewers #2 and #3. In our revision, we now include more injection site images in Figure 1B, and we include a new set of experiments to further validate our retrograde anatomy (Figures 1-S1 and S2), which also includes the injection sites from each animal. We do not think we have contamination in PL when targeting IL, as we see similar distributions when we target PL and IL in separate animals (Figure 1-S1). We also note that both of the retrograde experiments in Figure 1A-C are also substantiated by our anterograde experiments in Figures 1D-F. Together, we think these changes make our initial anatomy clearer and better motivate our other work.

2. The projections shown in Figure1E suggest that axons of both colors have strong presence in the IL and PL, and in Figure1-supplent 1C axons of both colors have strong presence in PL. Is this different from the authors interpretation? These data actually are more consistent with the conclusions in Hintiryan et al. that PL actually receives both inputs from rBLA and cBLA. This might also explain the detected responses in IL when stimulating rBLA: there was significant number of neurons responded, although smaller population. Also, it might be useful to add quantification in Figure1-supplent 1C, similar to Figure 1F.

The reviewer raises an interesting point and is correct that there are rBLA and cBLA axons in both PL and IL. While cBLA and rBLA provide different synaptic drive to PL and IL, there is quite a bit of axonal overlap, as described in Hintiryan et al. 2021. In our revision, we build on our initial axon anatomy in Figure 1 with new experiments in Figure1-S2 that shows how projections from PL- projecting rBLA neurons and IL-projecting cBLA neurons collateralize within the PFC. These findings motivated us to look at the physiological impact of these two BLA inputs to these two regions of the PFC. Using slice physiology, we first found these inputs target specific cell types in both PL and IL (Figures 2-3). Using in vivo physiology, we then found that rBLA and cBLA evoke different responses in PL and IL (Figure 4). Overall, we think our (brief) anatomical analysis in Figure 1 is generally in alignment with previous studies, such as seeing rBLA axon in both PL L2 and IL L5, which is similar to Hintiryan et al. 2021. However, we also think that our (much more thorough) and slice and in vivo physiology analysis in the other 6 figures provide a substantial advance over previous work. Importantly, we think our results argue that anatomy alone cannot show which networks are engaged, which is revealed using physiology.

3. There are strong presence of IL CA neurons located in L5 (Figure 3B) but their responses were not mentioned. According to the example images, it seems like CA neurons are predominantly located in the IL and PT neurons are predominantly located in the PL. This bias looks comparable if not stronger than the axonal preference in Figure 1E. How would this impact Figure 6 diagram?

The reviewer raises an important point, which was also noted by Reviewers #2 and #3, and which we now clarify in the text. We do not think that L5 PT cells are predominantly located in PL, as has been shown in multiple studies (Vertes, 2004; Gabbott *et al.*, 2005) and we have now chosen a more representative image. There is known to be a population of L5 CA neurons in IL with physiological properties and axonal projections that overlaps with L5 IT neurons (Avesar *et al.*, 2018; Gao *et al.*, 2022), which we have already shown are not targeted by cBLA. As suggested by the reviewers, we now include several new experiments to show that cBLA and rBLA do not target these neurons (Figure 2-S1) and that cBLA does not drive IL L5 CA and IL L2 CA neurons, even when IL L5 PT neurons are activated (Figure 5-S1). Note that we do not think that our summary diagram is impacted, as neither rBLA nor cBLA appreciably targets these neurons. Overall, we thank the reviewer for making this point, but we hope that our response clarifies any misunderstanding.

4. Are Figure 4 C and F essentially a reproduction of Figure 3C and D? If so, might consider move them to supplement figure.

We disagree with the reviewer that Figure 3C and 3D of our initial submission is reproduced in Figure 4C and 4F (note that these figures are now Figure 2 and 3, respectively). We would note that the first figure shows the “functional anatomy” of how rBLA and cBLA engage different cell types in PL and IL by restricting synaptic transmission to solely rBLA or cBLA terminals containing ChR2. In contrast, the latter figure shows the physiological responses, including the short-term dynamics, feed-forward inhibition, and evoked firing. These two data sets are complementary rather than overlapping, and together provide a comprehensive picture of how BLA engages PFC.

Reviewer #2 (Recommendations for the authors):This is a solid, well-designed study that provides a new level of anatomical and functional dissection of the BLA-mPFC circuit and is an important contribution to the field. The authors skillfully combined state-of-the-art techniques and the manuscript is well-written. My main criticisms are listed below and I would be happy to see the authors' response to these points:

We thank the reviewer for their very helpful comments, all of which we have addressed below.

• While the findings include a substantial amount of in vitro and in vivo electrophysiology, the potential behavioral roles of these distinct pathways remain mysterious. Is there any indication that stimuli with positive or negative valence differentially trigger activity in these mPFC regions, as would be expected from selective activation of the rBLA and cBLA in response to such stimuli? While such an experiment might be beyond the scope of the current manuscript, it would be good if the authors elaborate in the Discussion about the potential importance of this finding to value coding in the PFC.

The reviewer raises an interesting point, which we have now addressed via changes to the text. Based on previous literature, we agree that stimuli with positive and negative valence would engage different parts of BLA and thus regions of PFC. We agree with the reviewer that testing this idea is beyond the scope of the current manuscript, but we include an improved Discussion to address these ideas, including how better understanding of rBLA and cBLA connectivity to mPFC helps us better interpret previous studies of how mPFC encodes valence (page 15).

• The retrograde tracing in Figure 1 was done by injecting one AAVretro more anterior to the other. While it's understandable that the authors wished to avoid leak between the two sites, it seems important to verify the results (more rBLA cells labeled with the more rostral AAVretro injection) using single viral injections, or a set of mPFC injections done at the same AP location.

The reviewer makes an important point, and we now expand our retrograde anatomy by repeating our dual injections with single injections targeted at the same anterior-posterior coordinate. Encouragingly, we found nearly the same distribution of PL and IL projecting neurons across the rostral-caudal extend of the BLA (Figure1-S1). In separate experiments, we further validated the differential axon projections by injecting AAVrg-Cre in PL or IL, followed by AAV-DIO-YFP in the BLA. These experiments show a very similar distribution of axonal projections as in Figure 1D-F, further validating the differences in rBLA and cBLA connectivity, as well as our ability to target these regions with anterograde viruses. These results nicely complement our previous anatomical experiments, as well as those from other groups.

• The use of patterned blue light, combined with expression of either ChR2 or st-Chrome, needs to be better validated. Can the authors show that by using st-Chrome they are eliminating any direct responses to stimulation of cells in other layers? How were the conditions (light bar size, intensity) chosen for this experiment? This is an important calibration experiment and should be included in the manuscript.

The reviewer raises an important point. Please note we did these controls in Figure 6BandC, where patterned stimulation in L2/3 activated ChroME+ cells in L2/3 but not L5, and patterned stimulation in L5 activated cells in L5 but not L2/3, which allowed us to selectively activate L2/3. As described in the Methods, we chose conditions that allowed us to achieve selective activation of L2 CA neurons without activating L5 CA neurons at all. We would note that these details were included in the original submission, but we have now made them clearer.

• The recordings done using ChR2 (Figure 6S1) seem redundant with the st-Chrome recordings in Figure 6. Given the constraints of using a non-soma-targeted opsin in this experiment (which the authors nicely explain in the text), it might be best to remove this data or acquire similar data with st-Chrome.

We agree that the experiments in Figure 6-S1A-C are more informative for rBLA-evoked local circuity when using st-ChroME. We have now repeated these experiments in Figure 6D-E with st-ChroME, and find broadly similar results, with weaker excitation in L3 than L5 from L2 CA neurons.

• The authors state (p.18) that "cBLA does not participate in a reciprocal loop with IL, instead making unidirectional connections onto L5 PT neurons"; however, it's clear from the images (for example. Figure 6A) that the IL contains a fair amount of CA cells as well. Given the previous work by the Carter lab on reciprocal BLA-mPFC circuits, it would have been very interesting to see how these L5 CA cells compare with the PT cells in their inputs from the cBLA and from IL L2 CA cells.

The reviewer raises an important point, which was also raised by Reviewers #1 and #3. We suspect there may have been some confusion about the nature of CA neurons, which we now clarify in the text. There are indeed CA neurons in L5 of IL, but these are IT cells that also project to the contralateral PFC and striatum (Avesar *et al.*, 2018; Gao *et al.*, 2022). Therefore, we already started to examine the cBLA inputs onto these cells in our original manuscript, in which we showed that cBLA makes much stronger connections onto IL L5 PT cells than IL L5 IT cells. We now supplement these experiments with new monosynaptic synaptic targeting recordings, which show a strong bias from cBLA axons onto L5 PT neurons over L5 CA neurons in IL (Figure 2-S1). In our revision, we also perform new experiments testing whether IL L2 CA and IL L5 CA cells are activated by polysynaptic responses following cBLA stimulation of IL PT cells (Figure 5-S1). We find no evidence of cBLA activation of either IL L2 or IL L5 CA neurons, even when responses are suprathreshold at IL L5 PT neurons. We thank the reviewer for raising this point, which our new experiments have now addressed.

Reviewer #3 (Recommendations for the authors):Emerging evidence suggests the BLA are anatomically segregated based on molecular markers, axonal projection, and function. This paper further strengthened this view by identifying the rostral and caudal BLA (rBLA and cBLA) as a distinct subregion based on the projection and connection to prelimbic (PL) and infralimbic (IL) part of the PFC. Specifically, the paper demonstrated projection from rBLA and cBLA are connected most strongly to the L2 cortico-amygdala (CA) neurons in PL and L5 PT neurons in IL, respectively, and are capable of evoking spikes preferentially in these neurons/layers both in slice and in vivo. They also found little activity spread between IL and PL upon stimulation of r- or cBLA axons in each PFC subregion. Moreover, L2 CA neurons in PL are preferentially connected to the L5 PT neurons than L5 IT neurons locally. The data support the paper's main conclusion and illuminate the mechanistic underpinning of functional distinction within the BLA and PFC reported in earlier studies.The strength of the paper is the use of a wide variety of techniques to verify the subregion-specific connections between BLA and PFC and demonstrate this connectivity bias has functional relevance (i.e., causing subregion- and layer-specific activation of neurons with distinct local connectivity). No major weakness in technicality or data interpretation is noted.

We thank the reviewer for their enthusiastic critique, and address their specific points below.

1. The PL and IL (and ACC on this matter) are ambiguously defined. Please clarify how the authors distinguished them visually.

This is an important point, and we now better specify how we define these regions of PFC in the Methods. In our slice electrophysiology experiments, we defined PL and IL as 800-1200 µm and 1600-2000 µm from the dorsal surface. For our anatomy, we used the Allen Brain Atlas. Please note we have used similar definitions in several previous studies on these brain regions.

2. The retrograde tracer injection is targeted at L5 region of PL and IL (mediolateral 0.35 mm), despite in PL, it is not a major projection site for r- or cBLA. It will strengthen the paper if the degree of retrograde labeling bias in Figure 1C is shown to be preserved when tracer injection is targeted into the superficial layers, which appear to be the main projection site for both r- and cBLA (at least anatomically, as in Figure 1E).

The reviewer makes an important point, which was also raised by Reviewer #2. It is very difficult to restrict viral injections to superficial vs deep layers within PFC, as they are separated by only a few hundred microns, and we have not tried to do so in our study. However, we have now repeated these experiments with single AAVrg injections at the same anterior-posterior coordinate, which spread to both superficial and deep layers, and find consistent results to our earlier experiments (Figure 1-S1). We have also performed new anatomy experiments to assess the divergence of rBLA and cBLA inputs to PL and IL (Figure 1-S2). Lastly, we would note that superficial layers are not the main projection targets of cBLA, which sends axons to both L2 and L5 (Figure 1E), but mostly contacts L5 PT cells in L5 (as seen in Figures 2,3,4 and 5).

3. In Figure 3B, L5 PT neurons (labeled by retrograde tracer injection into PAG) is mostly in PL (and at the border between PL and IL). This makes me wonder how the approach reliably labeled L5 PT neurons in IL, and whether the PAG-projecting PT neurons are the representative subtype of PT neurons in IL. If this is not a representative image then I suggest replacing it with one that is.

The reviewer raises an important issue, which was also noted by Reviewers #1 and #2.

Although Figure 3B made it seem like L5 PT cells are mostly in PL, we see them in both PL and IL, as expected from the literature (Gabbott *et al.*, 2005). We have now replaced the image with a more representative image that better highlights the PT population in both PL and IL.

4. The latency of L5 PL spiking in vivo is similar between L2 CA and L5 PT neurons in PL and appears to be contrary to the polysynaptic circuit model (L2 CA to L5 PT) proposed to be engaged by rBLA. It would be helpful if the authors could discuss this discrepancy.

The reviewer makes an interesting point, which we address in the Discussion (page 12). The latency is not significantly different between L2/3 and L5 units, even though we expect L2 CA neurons to fire before L5 PT neurons in PL, but we do not view this as contrary to our results. Instead, we think it reflects both the number of activated units we were able to record in L2/3 and L5, and our inability to simultaneously record L2/3 and L5 units at the same time.

5. Color code for L2 CA IL and PL L5 PT is hard to distinguish (e.g., Figure 3C). I recommend using different color codes.

We have now adjusted the color scheme, and hope it will be easier to parse.

6. The vertical lines other than the Neuropixel tract are quite distracting in Figure 2. Supp 1. I suggest removing them if possible.

We have now expanded the size of the probe tract images to address this feedback.

7. Line 371: Figure S5 should be Figure 5

We have fixed this typo.

References

Avesar, D., Stephens, E.K. and Gulledge, A.T. (2018) Serotonergic Regulation of

Corticoamygdalar Neurons in the Mouse Prelimbic Cortex. *Front Neural Circuits*, 12, 63.

Gabbott, P.L., Warner, T.A., Jays, P.R., Salway, P. and Busby, S.J. (2005) Prefrontal cortex in the rat: projections to subcortical autonomic, motor, and limbic centers. *J Comp Neurol*, 492, 145-177.

Gao, L., Liu, S., Gou, L., Hu, Y., Liu, Y., Deng, L., Ma, D., Wang, H., Yang, Q., Chen, Z., Liu, D., Qiu, S., Wang, X., Wang, D., Wang, X., Ren, B., Liu, Q., Chen, T., Shi, X., Yao, H., Xu, C., Li, C.T., Sun, Y., Li, A., Luo, Q., Gong, H., Xu, N. and Yan, J. (2022) Single-neuron projectome of mouse prefrontal cortex. *Nat Neurosci*, 25, 515-529.

Vertes, R.P. (2004) Differential projections of the infralimbic and prelimbic cortex in the rat. *Synapse*, 51, 32-58.

Zhang, X., Guan, W., Yang, T., Furlan, A., Xiao, X., Yu, K., An, X., Galbavy, W., Ramakrishnan, C., Deisseroth, K., Ritola, K., Hantman, A., He, M., Josh Huang, Z. and Li, B. (2021) Genetically identified amygdala-striatal circuits for valence-specific behaviors. *Nat Neurosci*, 24, 1586-1600.

[Editors’ note: what follows is the authors’ response to the second round of review.]

The manuscript has been improved, but there are some remaining issues that need to be addressed, as outlined by reviewer 2 below:Note that all reviewers felt that the main strength of this manuscript is in the detailed synaptic physiology, whereas the anatomical identification of the cells as CA/PT is fuzzier and might inherently lump together multiple different functional classes

We appreciate this point, but note that the identification of CA and PT cells is not novel, rather building on considerable literature indicating they are distinct cell types. It is true that there are subtypes of L5 PT cells, which we have noted in the discussion. It is also possible there may be several L2 and/or L5 CA cells, but we have no evidence of this.

Reviewer #2 (Recommendations for the authors):Recommendation #1:Please add more detail regarding the injection sites (Figure 1 and Figure 1- S1 show one representative section in each animal). With the newly added soma sections in 1B, it will be interesting to state whether there are double-labeled neurons (magenta and cyan).

We have now added additional injection site images for the dual AAVrg-XFP experiments (see Figure 1 – S1H and I) and have included more information about the injection site locations in the figure legends. Please note that we have already quantified the amount of overlap between IL and PL projecting, dual-labeled cells in the BLA and found relatively few of them across the R-C axis (see Figure 1C).

Related, it will be good to explain in Figure1-S2 B experiment, whether the injection of AAVrg-cre mostly resides in L2 (Similar experiments in Figure 1-S2F nicely show that injection covers the entire IL). This will strengthen the observation that rBLA mainly projects to L2 in PL, and this is not due to that the injection only targets L2.

Please note that we have already stated that the AAVrg-Cre injection does not reside in a specific layer. In general, we have found it is almost impossible to restrict AAVs to particular layers, and AAVrg certainly spreads too far (typical radius ~500 µm). In our revision, we have now added a plot of fluorescence vs laminar location of the PFC for all of the injections (Figure 1 – S2C and H). We found that both our IL and PL targeted injections of AAVrg-cre span L2 (180-220 µm from pia) and L5 (350-650 µm).

Recommendation #2:Please add quantification in Figure 2A image for layer distribution which will help readers appreciate the cell distribution of the projection neurons, and also cite Ferreira et al., 2015 paper which has the same PAG and amygdala injection and quantified the distribution.

As requested, we have now further quantified the distributions of cells across layer (Figure 2 – S1J). Please note that this kind of analysis has already been done in several previous papers, many of which we previously cited, and we have now also added the Ferreira et al., 2015 paper, as requested.